



# Surges of Harald Moltke Bræ, north-west Greenland: Seasonal modulation and initiation at the terminus

Lukas Müller[1,2], Martin Horwath[1], Mirko Scheinert[1], Christoph Mayer[3], Benjamin Ebermann[1], Dana Floricioiu[4], Lukas Krieger[4], Ralf Rosenau[1], and Saurabh Vijay[5,6]

[1]Technische Universität Dresden, Institut für Planetare Geodäsie, Dresden, Germany
[2]ETH Zurich, Institute of Geodesy and Photogrammetry, Zurich, Switzerland
[3]Bavarian Academy of Sciences and Humanities, Munich, Germany
[4]German Aerospace Center, Wessling, Germany
[5]DTU Space - National Space Institute, Kongens Lyngby, Denmark
[6]Byrd Polar & Climate Research Center, Columbus, USA

**Correspondence:** Lukas Müller (lukamueller@ethz.ch)

**Abstract.**

Harald Moltke Bræ, a marine-terminating glacier in north-west Greenland, shows episodic surges. A recent surge from 2013 to 2019 lasted significantly longer (6 years) than previously observed surges (2-4 years) and exhibits a pronounced seasonality with flow velocities varying by one order of magnitude (between about $0.5$ and $10\,\mathrm{m/day}$) in the course of a year. During this six-year period, the velocity always peaked in the early melt season and decreased abruptly when meltwater runoff was maximum. Our data suggest that the seasonality has been similar during previous surges, and, to a much lesser extent, during the intermediate quiescent phases. It is peculiar to Harald Moltke Bræ that the seasonal amplitude is amplified episodically to constitute glacier surges. The surge from 2013 to 2019 was preceded by a rapid frontal retreat and a pronounced thinning at the glacier front ($30\,\mathrm{m}$ within 3 years).

We discuss possible causal mechanisms of the seasonally modulated surge behaviour by involving various system inherent factors (e.g. glacier geometry) and external factors (e.g. surface mass balance). The seasonality may be caused by a transition of an inefficient subglacial system to an efficient one, as known for many glaciers in Greenland. The patterns of flow velocity and ice thickness variations indicate that the surges are initiated at the terminus and develop through an up-glacier propagation of ice flow acceleration. Possibly, this is facilitated by a simultaneous up-glacier spreading of surface crevasses and weakening of subglacial till. Once a large part of the ablation zone has accelerated, conditions may favour substantial seasonal flow acceleration through seasonally changing meltwater availability. Thus, the seasonal amplitude remains high for two or more years until the fast ice flow has flattened the ice surface and the glacier stabilizes again.

## 1 Introduction

Recent optical and radar remote sensing observations provide unprecedented insights into the dynamics of ice sheets and glaciers (Moon et al., 2014; Vijay et al., 2019). We applied such data to investigate long-term ice dynamics of Harald Moltke Bræ, a marine-terminating outlet glacier in north-west Greenland. Fig. 1 shows the surface ice flow velocity over the past 23



years derived from Landsat (Rosenau, 2014), Sentinel-1 (Solgaard and Kusk, 2019) and TerraSAR-X (Joughin et al., 2020) at a point (Fig. 2) close to the contemporaneous terminus position of Harald Moltke Bræ. This time series confirms previously documented phases of significantly accelerated ice flow in 1999/2000 (Joughin et al., 2010; Rosenau, 2014), 2005/2006

(Rosenau, 2014) and 2013/2014 (Hill et al., 2018). Additionally, it shows pronounced seasonal variations from 2013 to 2019 with velocities varying by one order of magnitude in the course of a year. At the end of 2019 and the beginning of 2020, the velocities remained low and no significant variations were observed.

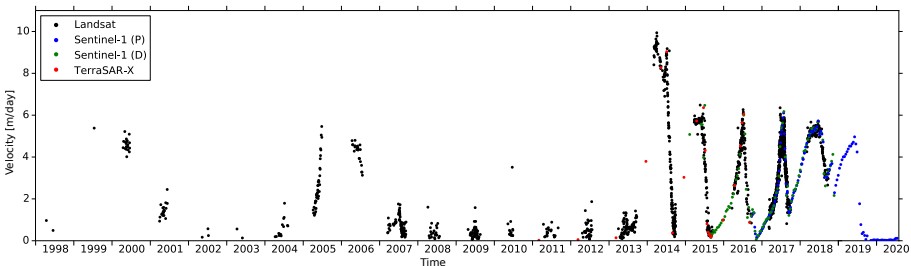

**Figure 1.** Flow velocity derived from Landsat, Sentinel-1, and TerraSAR-X at a point close to the terminus of Harald Moltke Bræ. For Sentinel-1 we use two different velocity data sets: one processed at Denmark's National Space Institute (DTU Space), referred to as Sentinel-1 (D), and another one provided by the Programme for Monitoring of the Greenland Ice Sheet (PROMICE), referred to as Sentinel-1 (P).

Harald Moltke Bræ is located in north-west Greenland at about $76.5\,°\mathrm{N}$ (Fig. 2). It is one of three glaciers that terminate in the Wolstenholme Fjord (Mock, 1966). At its present (2020) front the glacier is about 5 km wide. Starting at the present

terminus its main stream can be tracked about 65 km upstream (Fig. 2, red line). At a distance of about 10 km from the present glacier front the Blue Ice Valley Glacier (Mock, 1966) flows into Harald Moltke Bræ. The boundary between these two streams is clearly visible by a medial moraine along the entire distance from their confluence to the terminus. Based on the Arctic Digital Elevation Model (ArcticDEM) and flow lines identified in Landsat images, we estimate the size of the overall drainage basin to be about $1500\,\mathrm{km^2}$ consisting of Harald Moltke Brae ($1200\,\mathrm{km^2}$) and Blue Ice Valley Glacier ($300\,\mathrm{km^2}$). As

another remarkable feature, a $3\,\mathrm{km}$ long and $1\,\mathrm{km}$ wide lake abuts to the northern side of Harald Moltke Bræ at about $20\,\mathrm{km}$. It might be an additional source of freshwater influx into the glacier system.

Already prior to the era of satellite remote sensing, significant changes in the dynamic behaviour of Harald Moltke Bræ were reported. Wright (1939) observed an exceptional advance of the glacier front by about $2\,\mathrm{km}$ between 1926 and 1928 and inferred that the average surface flow velocity in this interval was at least $1000\,\mathrm{m/year}$ ($2.7\,\mathrm{m/day}$). Mock (1966) used

the displacement of ice-surface features visible in aerial and terrestrial photographs to show that between 1954 and 1956 the average velocity was about $1\,\mathrm{m/day}$, ten times higher than the average velocity between 1937 and 1938.

Based on its periods of clearly accelerated ice flow, Harald Moltke Bræ has been assumed to be a surge-type glacier (Rignot et al., 2001; Rosenau et al., 2015; Joughin et al., 2010; Hill et al., 2017, 2018). In general, the flow behaviour of a surge-type glacier is characterized by an alternation of long periods of low velocity (3-100 years, quiescent phases) and comparably short

periods with velocities increased by at least one order of magnitude (1-10 years, active phases or surge events) (Jiskoot, 1999;





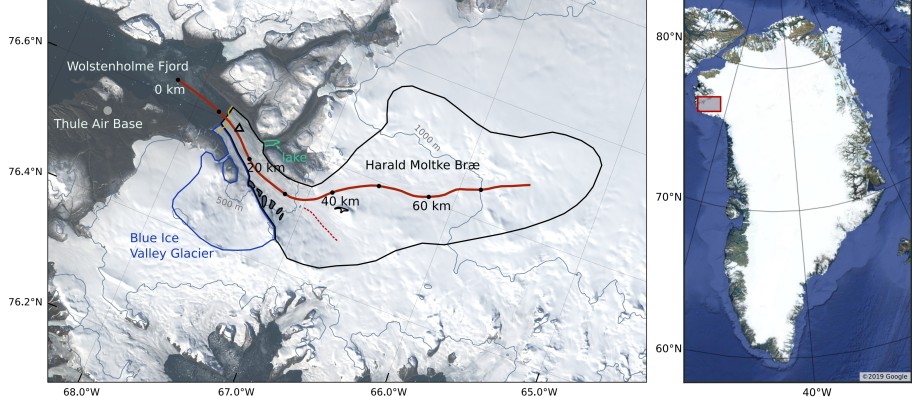

**Figure 2.** Region of investigation as viewed in a Landsat 8 scene of August 2014 (USGS, 2019) (see red box in right image for its location in Greenland). Blue and black lines mark the drainage basins of Blue Ice Valley Glacier and Harald Moltke Bræ, respectively. Red lines (solid and dashed) mark the approximate center line of the main glacier stream and of a tributary, respectively. The starting point of the center line is at the mean position of the glacier front line in 1916 (year of first available record of front position). We use the glacier front position of 1916 as a reference for all longitudinal profiles and glacier front positions. The yellow line marks the approximate front positions of 2020. The black triangle marks the point (67.628°W, 76.588°N) to which all velocity time series in this paper refer.

Benn and Evans, 1998). During a quiescent phase, the lower part of the glacier flows slower than the upper part. Therefore, the transition part, called reservoir area, thickens dynamically while the lower part, called receiving area, thins (Jiskoot, 1999). This pattern reverses in the active phase. When ice flow in the receiving area is enhanced during the active phase the glacier front advances rapidly. Most known glacier surges started with an increase of flow velocities in the upper part of the glacier
and propagated down-glacier (Raymond, 1987; Solgaard et al., 2020; Wendt et al., 2017). However, some glaciers, such as the Aventsmarksbrae in Svalbard (Sevestre et al., 2018), exhibit an opposite sequence with an initiation of the surge at the glacier front and its subsequent propagation up-glacier.

Only a small proportion of about 1 % of the glaciers worldwide is assumed to exhibit a surge-type behaviour (Jiskoot, 1999). Such glaciers normally cluster in certain regions (such as Alaska, Svalbard and the Karakoram) where the conditions
are favourable for a surge behaviour (e.g. a soft glacier bed and an at least partially temperate regime) (Benn et al., 2019). Some surge glaciers are also known in Greenland, such as the Hagen Bræ in north-east Greenland (Solgaard et al., 2020). In north-west Greenland, however, no surge glaciers are known except for Harald Moltke Bræ (Hill et al., 2017).

Two different mechanisms, both involving an internal instability, have been considered as possible causes of glacier surges: (A) thermal and (B) hydrological (Murray et al., 2003). (A) The base of a polythermal glacier can be partly frozen to the glacier
bed such that the ice flow is in part restricted during the quiescent phase. The increasing shear stress due to the steepening of the reservoir area during the quiescent phase can subsequently trigger a feedback with an initially slow movement producing friction heat. This leads to a further acceleration and enables basal sliding over large parts of the glacier base (Murray et al., 2003; Jiskoot, 1999). (B) A hydrologically driven surge event occurs when the subglacial drainage system closes after having



been efficient over several years (quiescent phase), and becomes inefficient. When an increasing amount of meltwater meets
this inefficient drainage system, subglacial water pressure increases and induces basal sliding (Murray et al., 2003; Jiskoot,
1999). Previous studies tended to explain the surge behaviour of Harald Moltke Bræ by a feedback mechanism associated with
weakening of soft subglacial sediments, that is, by a thermally driven mechanism (Hill et al., 2018; Joughin et al., 2010; Mock,
1966).

While system inherent drivers are responsible for cyclic surge behaviour, the seasonality of the flow velocities is mostly
ascribed to external forcing changing in the course of a year. According to Moon et al. (2014) seasonal velocity variations of
a large number of Greenlandic glaciers result from either the seasonally changing ice front position or the varying meltwater
availability.

In this paper, we focus on the three active phases of Harald Moltke Bræ detected by satellite remote sensing, 1999-2000,
2005-2006 and 2013-2019. The observation of the surge 2013-2019 is unprecedented in terms of the simultaneous strong
seasonality with velocities decreasing to the level of the quiescent phase always in summer. In addition, the active phase
2013-2019 is significantly longer (6 years) than the previous two active phases, 1999-2000 and 2005-2006 (always 2 years).
In contrast to most known surge glaciers, the surges at Harald Moltke Bræ are shown to be initiated at the glacier front and
to develop up-glacier. We analyse and interpret this extraordinary flow behaviour in terms of possible causal processes. For
this, we correlate spatio-temporal flow patterns to several system-inherent parameters (e.g. glacier geometry) and external
parameters (e.g. surface mass balance). We support the interpretation by analysing dynamic ice-height changes predicted from
observed velocity variations.

## 2 Data and Methods

### 2.1 Ice flow velocity data sets

To determine flow velocity fields for outlet glaciers in Greenland, Rosenau et al. (2015) developed a processing scheme based
on the feature tracking method using suited Landsat images available since 1972. Resulting velocity fields are used in the
present study. In addition, we included three velocity datasets derived from SAR offset tracking: Sentinel-1 (P) and Sentinel-1
(D) processed by PROMICE (Solgaard and Kusk, 2019) and DTU Space, respectively, and TerraSAR-X provided by MEa-
SUREs (Joughin et al., 2020). Tab. 1 provides technical details about these four data sets. All are combined to derive a time
series of monthly averaged velocity fields with a spatial and temporal coverage as high as possible (Appendix A). Thereby, an
almost seamless time series can be inferred for the period after 2013. Gaps in the Landsat data set caused by polar night are
filled by data from SAR based techniques. The accuracy of the joint velocity time series is shown to be better than $0.5\,\mathrm{m/day}$
over most of the time, but can exceed $1\,\mathrm{m/day}$ in single months with rapidly changing ice flow (Appendix A).





**Table 1.** Overview over the technical details of the velocity data sets from Landsat, Sentinel-1 (D), Sentinel-1 (P) and TerraSAR-X. The velocity fields are given in form of regular spatial grids. The time basis refers to the acquisition time difference of the image pairs used for the velocity determination. The temporal resolution is the time difference between consecutive velocity fields.

| Data set | Landsat | Sentinel-1 (D) | Sentinel-1 (P) | TerraSAR-X |
|---|---|---|---|---|
| time span | 1998-2018 | 2014-2018 | 2016-2020 | 2011-2017 |
| spatial resolution [m] | 150-600 | 300 | 500 | 100 |
| time basis [days] | 5-100 | 12 | 12 | 11 |
| temporal resolution [days] | few days | 12 | 6/12 | 11-351 |
| source | TU Dresden | DTU Space | PROMICE | NSIDC |

## 2.2 Ice front position

For the period between 1916 and 1960 ice front positions are taken from the previous studies (Koch, 1928; Wright, 1939; Davies and Krinsley, 1962). For the period after 1972, ice front positions are digitized on the basis of Landsat images. The variation of the front position is measured by the average distance from its position in 1916 applying a method proposed by Moon et al. (2008) (Appendix B).

## 2.3 Surface topography

We use four different digital elevation models (DEMs): the ArcticDEM (June 2018) and two interferometric DEMs calculated from repeated TanDEM-X (TDM) acquisitions in January 2011 (2011-01-07, 2011-01-13, 2011-01-18, 2011-01-24) and December/January 2013/2014 (2013-12-16, 2013-12-22, 2014-01-13) (Krieger et al., 2020). The interferometric DEMs from TanDEM-X have been vertically co-registered over flat, ice free terrain adjacent to the Wolstenholme Fjord by adjusting their absolute phase offset (Krieger et al., 2020). Ice-surface height-change rates for the intermediate periods are obtained by computing the differences between these DEMs.

In addition to the measured surface elevation changes, we computed the monthly dynamic ice-height change rates to be expected due to the flow velocity distribution. This provides information on geometric glacier changes with a high sampling rate, and enables a better understanding of how these changes are related to the flow velocity changes. To do so, we consider only the horizontal components of ice flow and assume parallel ice flow as well as a constant density of the glacier ice. Then, the relationship between ice flow and the dynamic height change at a given point is

$$\frac{\partial H}{\partial t} = -v \cdot \frac{\partial H}{\partial x} - H \cdot \frac{\partial v}{\partial x}. \tag{1}$$

$H$ denotes the glacier thickness, $t$ denotes time, $x$ denotes the position along the flow line and $v$ is the velocity averaged over the vertical column of the glacier. We evaluate $v$ by multiplying the observed surface velocities with a factor of $0.9$ This factor adopted in the absence of more specific information is a rough approximation and a potential source of error particularly





for surge-type glaciers (Wu and Jezek, 2004; Andersen et al., 2015). Note that the total surface height change sums up from
this dynamic height change and the height change due to surface mass balance (SMB).

We implement Eq. 1 by using velocity fields and ice thickness data (see next section) interpolated to the main flow line. To
suppress noise, we subsequently fit a 3rd degree polynomial to the results.

## 2.4 Bedrock topography

BedMachine (Morlighem et al., 2017) provides both ocean floor and bedrock topography in a gridded format. Additional
bedrock data are available along profiles of airborne ground penetrating radar measurements by the Center for Remote Sensing
of Ice Sheets (CReSIS). Based on a comparative analysis of BedMachine and CreSIS data (Section 3.7 and Appendix C) we
use BedMachine data only for areas above $16\,km$.

## 2.5 Parameters of external forcing

The Regional Climate Model (RACMO2.3p2) provides surface mass balance (SMB), air temperature, precipitation and snow
melt on a monthly basis. We infer the monthly mean SMB, precipitation and snow melt averaged over the overall drainage
basin.

Additionally, we compute the ice-mass flux through a cross section orthogonal to the flow lines located close to the glacier
front at about $18\,km$. To determine the cross-sectional area, we use the ArcticDEM and BedMachine.

## 2.6 Visible features

We also assess changes traceable by visual inspection of the Landsat scenes. We focus on four different features: lakes on
the glacier surface, outflow of subglacial meltwater at the glacier front (meltwater plumes), sea ice coverage in the fjord, and
calving events (Appendix E). We assess these features by distinguishing between three states: not visible, moderate and strong
occurrence.

# 3 Characterization of spatio-temporal patterns of glacier geometry, flow velocity and external influences

## 3.1 Front position

Fig. 3 shows that most of the active phases between 1916 and 2020 were accompanied by an advance of the terminus inter-
rupting its long-term retreat (Mock, 1966). From 1926 to 1928 the frontal advance was more pronounced than in later active
phases. During the surge event in 1954-1956, the glacier front retreated slightly.

Besides documented surge events, there are further phases of frontal advance between 1970 and 1985 for which, however,
independent velocity observations are not available. Further, Fig. 3 indicates a long-term acceleration of the frontal retreat
rate from about $100\,m/year$ before 2000 to about $200\,m/year$ thereafter (approx. $8\,km$ between 1920 and 2000 compared to
approx. $4\,km$ between 2000 and 2020).

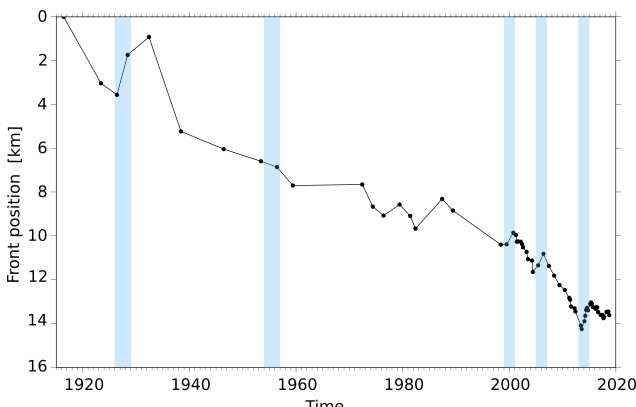

**Figure 3.** Average front position with respect to the terminus of 1916 consistent with the profile line shown in Fig. 2. The surges are marked by light blue color.

## 3.2 Ice flow during the active phases 1999/2000 and 2005/2006

In both active phases, 1999-2000 and 2005-2006, the glacier exhibits high velocities ($> 4\,\mathrm{m/day}$) in late spring (Fig. 4). There
is a rapid acceleration in spring 2005 and a rapid slowdown in summer 2006. In the years 2001, 2004 and 2007, which are before or after surge periods, an acceleration in spring and peak velocities in early summer are also visible. Over most of the rest of the quiescent phases the flow velocities remain clearly below $1\,\mathrm{m/year}$ with no significant fluctuations.

During the quiescent phases the glacier front retreats (Fig. 3,4), indicating that the flow velocity at the front is lower than the calving rate. In the years before the surge 2005-2006, this retreat was about $500\,\mathrm{m/year}$. In the active phases, the glacier front
advances by about $500\,\mathrm{m/year}$, indicating that the flow velocity exceeds the calving rate.

Generally, no clear correlation between interannual variations of flow velocities and external influences (precipitation, runoff and SMB) could be identified (Fig. 4). Due to exceptional high precipitation the SMB peaks in 2004 shortly before the initiation of the surge in spring 2005. The basin average annual SMB is negative after the termination of the surge in 2006 due to increased meltwater runoff. This implies that after 2006 the glacier looses mass even without ice discharge by calving.

The velocity fields visualized in Fig. 5 show that in both active phases, 1999-2000 and 2005-2006, the velocities were highest at the glacier front and decreased approximately linearly with increasing distance from the front. As indicated in Fig. 5c and 5d, shortly after the surge initiation (July 1999 and June 2005) fast ice flow is found especially in the lower part of the glacier associated with steeply sloped velocity profiles. Towards the end of a surge (e.g. June 2000 and July 2006), however, upper parts of the glacier were increasingly affected by fast ice flow, whereas the velocities were decreasing close to the terminus.

## 3.3 Ice flow during the active phase 2013-2019

In autumn/winter 2013 there was an abrupt change from constantly low velocities of less than $1\,\mathrm{m/day}$ in all months with available data to pronounced seasonal fluctuations over an order of magnitude with maximum velocities of $6-10\,\mathrm{m/day}$





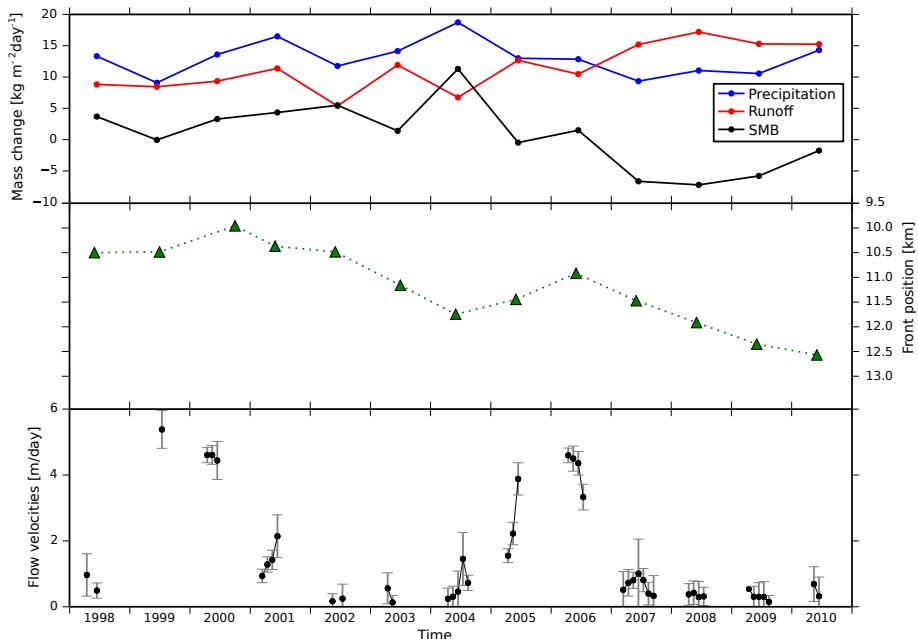

**Figure 4.** Bottom: Monthly glacier velocity and its uncertainty estimate derived from the Landsat data set at a point close to the terminus of Harald Moltke Bræ (Black triangle in Fig. 1). Middle: Monthly front position with respect to the year 1916 digitized in Landsat images where available. Top: Yearly precipitation, meltwater runoff and SMB (difference between precipitation and runoff) averaged over the drainage basin.

in spring and summer (Fig. 6). Whereas the velocities dropped well below $1\,\mathrm{m/day}$ in summer 2014, 2015 and 2016, they remained on a medium level above $1\,\mathrm{m/day}$ in summer 2017 and 2018.

The front retreated by $1000\,\mathrm{m/year}$ between 2011 and 2013 (Fig. 6), faster than in the previous quiescent phases ($500\,\mathrm{m/year}$), possibly due to a long-term increase of the calving rate. Due to this frontal retreat the constant point to which the velocities in Fig. 6 refer is located closer to the front in 2013. We note that this closer location to the glacier front could be one reason for the higher maximum velocity in 2013 compared to previous surges. However, it cannot be the main cause for the increase of flow velocities by more than one magnitude in 2013, since the acceleration in 2013/2014 affects the entire part between 13

and $25\,\mathrm{km}$ (Fig. 7). The high average velocity between 2013 and 2015 exceeded the calving rate and led to a rapid advance of the glacier front. Thereafter the terminus retreated on average by $200\,\mathrm{m/year}$, since the rather short-term accelerations in 2016 and 2017 were not sufficient to compensate the calving rate.

The average SMB was exceptionally high ($10\,\mathrm{kg\,m^{-2}\,day^{-1}}$) in 2013, the year of surge initiation. In contrast to the SMB maximum in 2004, the SMB maximum in 2013 is due to a low meltwater runoff rather than a high accumulation rate.

Directly before (September 2013) and after a phase of accelerated ice flow (August 2014), the velocities were about $1\,\mathrm{m/day}$ or less over almost the entire glacier (Fig. 7). A slight acceleration already occurred in July and August 2013 (Fig. 6). From September to December 2013 the velocities increased rapidly in the part below $20\,\mathrm{km}$, whereas the parts further upstream were

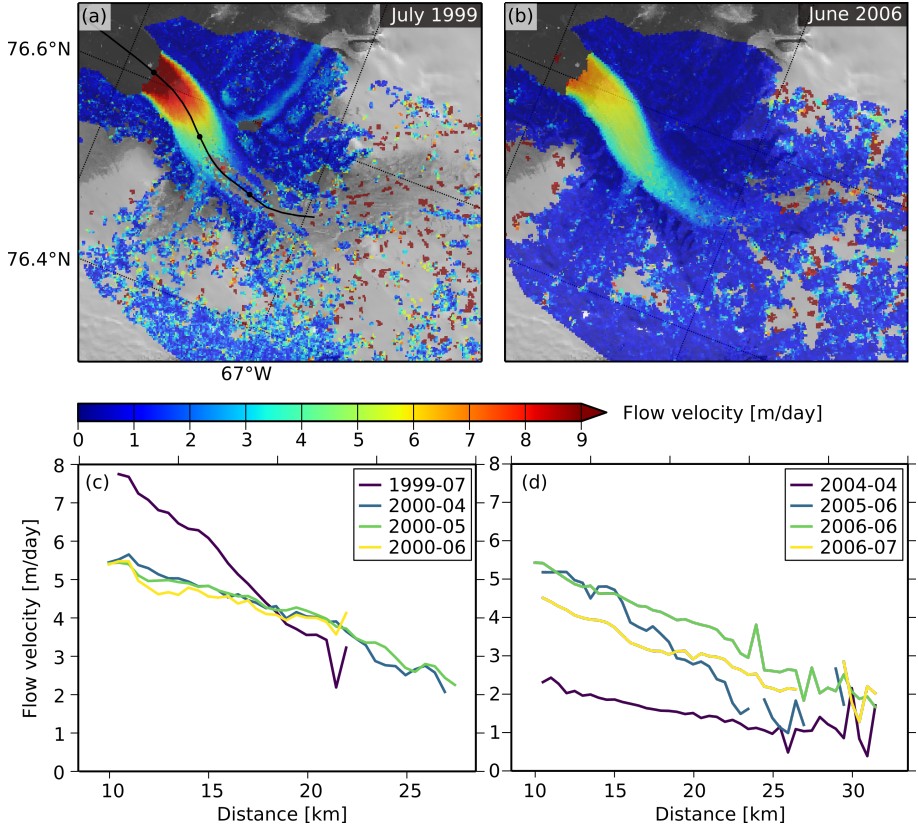

**Figure 5.** Exemplary velocity fields (a and b) and velocity profiles (c and d) for the surges 1999/2000 and 2005/2006. The profile location is shown as a black line in (a), with black circles every $10\,\mathrm{km}$ along the profile. The background image in (a) and (b) is a Landsat 8 scene (USGS, 2019).

less affected or not affected at all. Subsequently, also the upper parts of the glacier accelerated significantly leading to velocities of more than $5\,\mathrm{m/day}$ within most of the glacier area below $25\,\mathrm{km}$.

The longitudinal profiles in Fig. 7e show that a small part at the glacier front with an extension of $2-3\,\mathrm{km}$ had slightly increased velocities already in September 2013. Between April and July 2014, the ice flow was slowing down from month to month between 13 and $22\,\mathrm{km}$, whereas it was continuing to accelerate further upstream. After the slowdown between July and August 2014, the glacier had lowest velocities at the glacier front and a velocity maximum at about $29\,\mathrm{km}$, unlike September 2013.

The cross velocity profiles in Fig. 7f show that both streams, the main stream of Harald Moltke Bræ and the part of Blue Ice Valley Glacier, were equally affected by the surge. Thus, the velocity changes had a uniform effect over almost the entire width of the glacier.

  Two distinct patterns of seasonal ice flow variations can be identified in Fig. 8a. In 2014, 2015 and 2018, the glacier reached high velocities of more than $4\,\mathrm{m/day}$ already between January and March. By contrast, in 2016 and 2017, the velocities





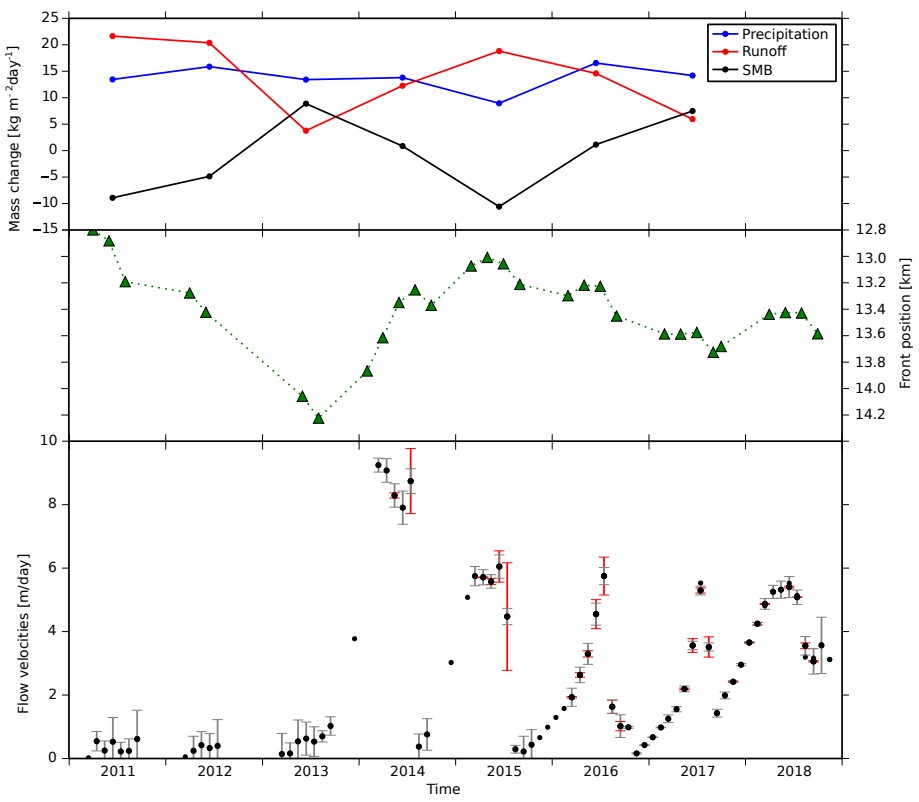

**Figure 6.** Velocities and parameters like in Fig. 4, but for the period 2011-2018. Additionally, red error bars indicate the standard deviations of monthly velocity values from different data sets.

remained below $2\,\mathrm{m/day}$ in autumn and increased only gently at the beginning of the year followed by a rapid acceleration between May and June. The rapid slowdown in summer around July and August is common to all years between 2013 and 2018. These decelerations coincide with the maximum of the meltwater runoff (Fig. 8b).

The longitudinal velocity profiles for July of the years 2015 to 2018 (Fig. 9a) all show a similar decrease from about $6.5\,\mathrm{m/day}$ at $14\,\mathrm{km}$ to about $2\,\mathrm{m/day}$ at $31\,\mathrm{km}$.

In contrast to July, the profiles for September vary largely over the years 2015 to 2018 (Fig. 9b). In 2015 and 2016, velocities remained relatively low and had their maximum at approximately $29\,\mathrm{km}$. In 2017 and 2018 (years with rapidly increasing velocities in autumn), velocities were highest at the glacier front. In 2015, 2016 and 2017, ice flow was enhanced within $2-3\,\mathrm{km}$ from the terminus.

### 3.4 Ice flow during the quiescent phases

In the quiescent phase, velocity maxima amounted to $0.3-0.4\,\mathrm{m/day}$ both in the main stream of Harald Moltke Bræ (about $29\,\mathrm{km}$) and in the Blue Ice Valley Glacier about $2\,\mathrm{km}$ upstream from the confluence with Harald Moltke Bræ (Fig. 10).



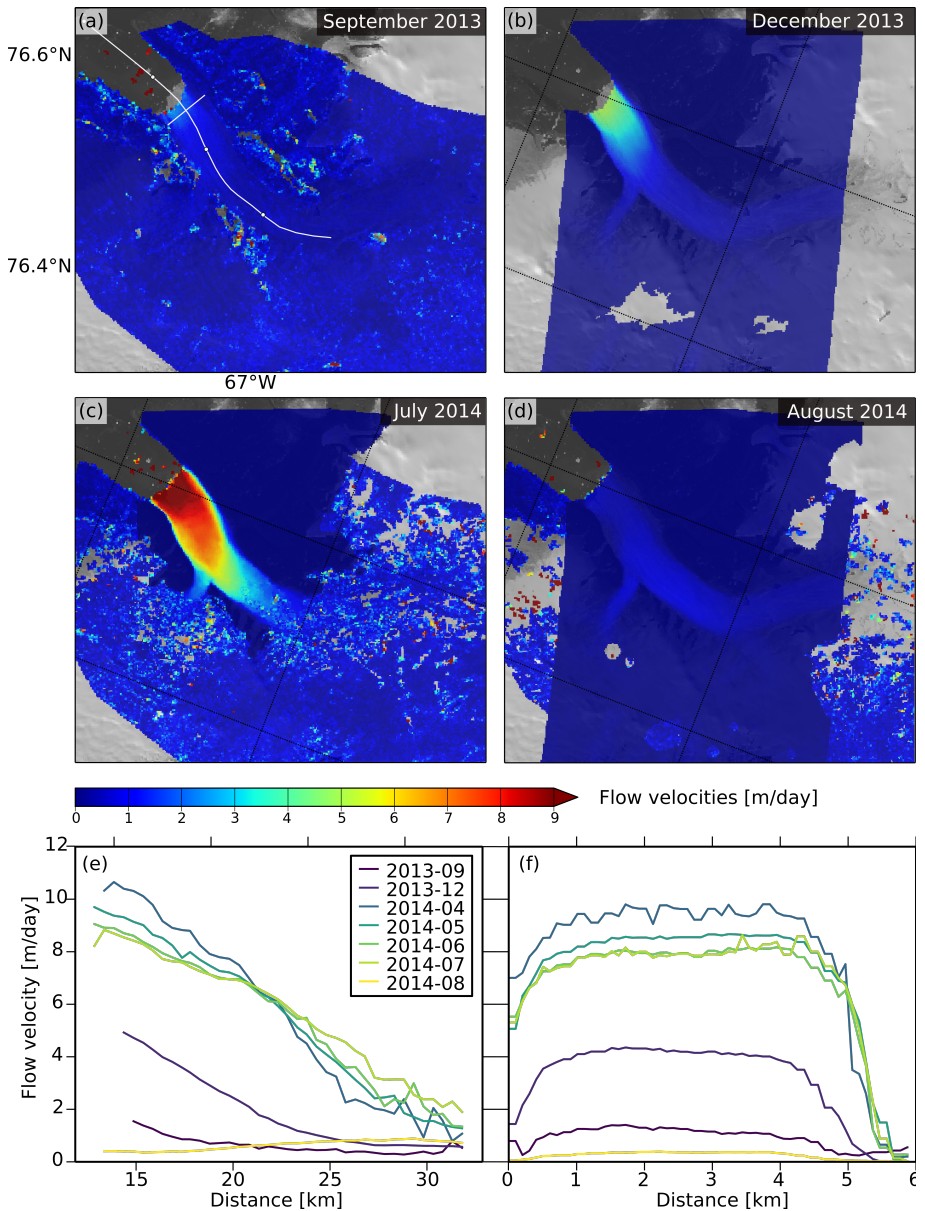

**Figure 7.** (a)-(d): Velocity fields for four selected months within the period from September 2013 to August 2014. The background image is a Landsat 8 scene (USGS, 2019). (e) and (f): Longitudinal and cross velocity profiles for the same year. Profile locations are shown in (a) (longitudinal profile is the same as in Fig. 5).

Velocities were lowest (lower than $0.2\,\mathrm{m/day}$) within an area of about $8\,\mathrm{km}$ extent right below the confluence. Within 2-3 km from the terminus, however, velocities were slightly higher at about $0.3\,\mathrm{m/day}$ and $0.5\,\mathrm{m/day}$ in March 2012 and March 2013, respectively.



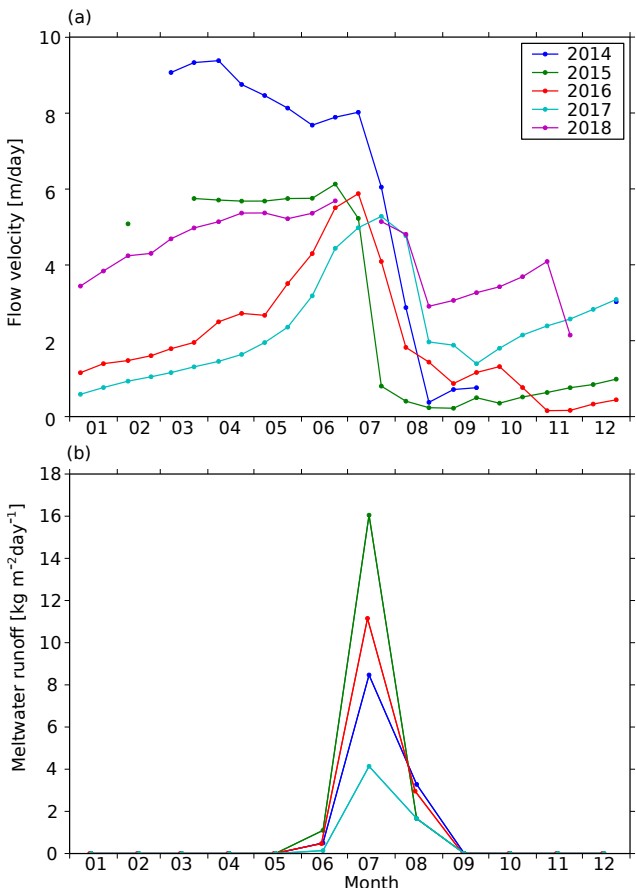

**Figure 8.** Seasonality of flow velocities (semimonthly velocities at the point marked by the black triangle in Fig. 1) (a) and meltwater runoff (b). For 2018, data of meltwater runoff were not available.

## 3.5 Visually inspected features

In years with high flow velocities (2014-2018) both calving events and the sea ice break-up in the fjord started earlier compared to the preceding quiescent phase (Fig. 11). Meltwater plumes were larger in the active phase 2013-2019 than between 2011 and 2013. However, the extent of meltwater lakes on the glacier surface was about the same before and after 2013.

Landsat images did not show any significant changes in the lake adjacent to the northern side of the glacier.

The middle moraine between Blue Ice Valley Glacier and the main stream of Harald Moltke Bræ, as inspected from the Landsat images, does not show any significant change of position or surge looped geometry (Appendix E), even during phases of highly variable flow velocities. This indicates that both glaciers were identically involved in the surge.


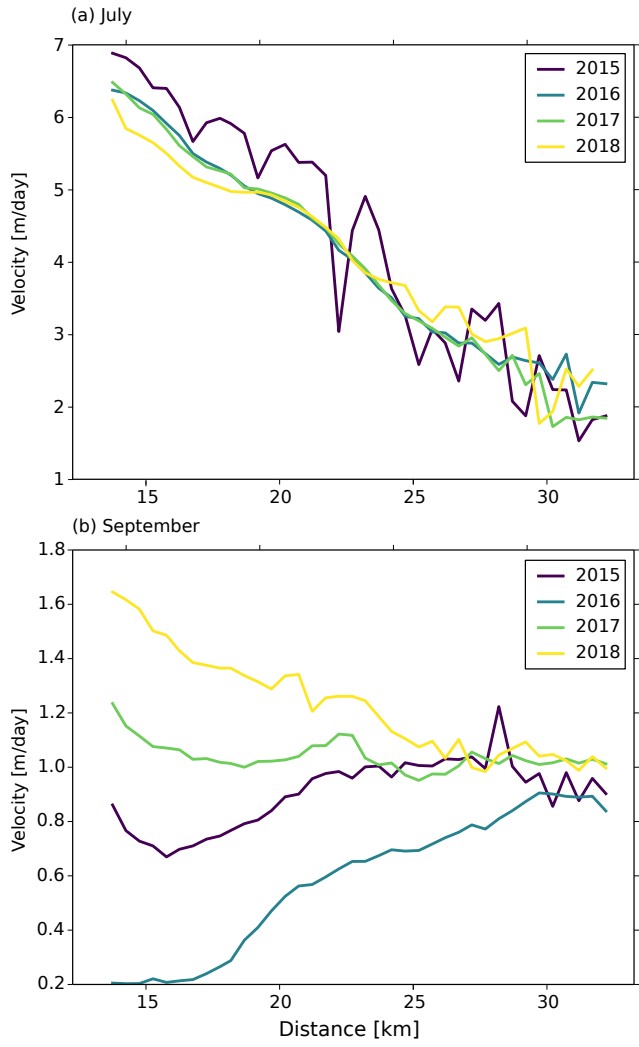

**Figure 9.** Longitudinal velocity profiles for always a month with high velocities (July) (a) and a month with low velocities (September) (b).

## 3.6 Ice-mass balance

The time series of the mass flow through a cross-sectional area close to the glacier front (Fig. 12) has three clear steps marking
215 the surge events. During the surge phases, the estimated average ice mass discharge was $0.5-1\,\mathrm{Gt/year}$ subject to uncertainties
associated to the interpolation of flow velocities to unobserved months. During the quiescent phases, mass flux was mostly
below $0.05\,\mathrm{Gt/year}$. The average SMB was roughly $0.2\,\mathrm{Gt/year}$ in 1990-1998 and $0.1\,\mathrm{Gt/year}$ in 2001-2004. Thus, between
1998 and 2006, the ice discharge exceeded SMB during the active phases, while SMB exceeded discharge during the quiescent
phases. After 2006, however, both the ice flux and SMB contribute to an overall mass loss of about $0.4\,\mathrm{Gt/year}$. SMB maxima
in 2004 and between autumn 2012 and spring 2014 precede the active phases 2005-2006 and 2013-2019.



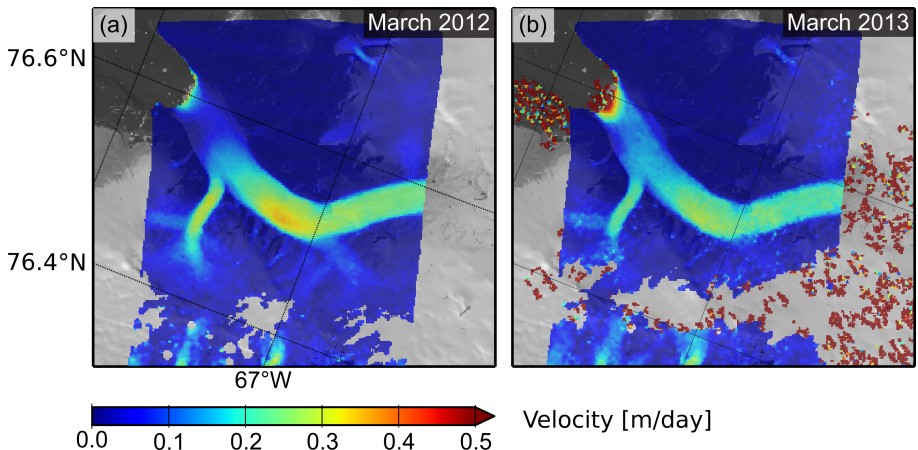

**Figure 10.** Exemplary velocity fields during a quiescent phase in March 2012 and 2013. The background image is a Landsat 8 scene (USGS, 2019). The color scale is different from that in the previous figures.

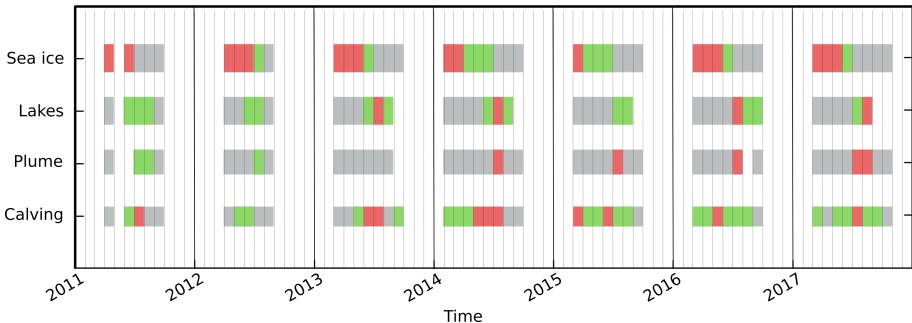

**Figure 11.** Occurrence of four different visual features between 2011 and 2018. Grey: not visible; Green: moderate; Red: strong.

### 3.7 Bedrock and ice surface geometry

We examined the bedrock topography from BedMachine (Morlighem et al., 2017) and from CReSIS, both along the same CReSIS flight path (Fig. 13). BedMachine suggests a trough at $10-12\,\text{km}$ (close to the front position of the year 2000) and a ridge at about $13\,\text{km}$ (close to the front position in 2015) associated with an ice thickness as low as $40\,\text{m}$. The BedMachine uncertainty in this section is specified at a high level of $100\,\text{m}$. From $13\,\text{km}$ up-glacier the BedMachine bedrock declines steeply to a depth of $200\,\text{m}$ at $16\,\text{km}$ with an uncertainty of $10-20\,\text{m}$ (Morlighem et al., 2017). However, between 13 and $16\,\text{km}$, CReSIS data deviate significantly from BedMachine and suggest a gently sloped bedrock with a depth of about $230\,\text{km}$ at the front position of the year 2015. From 16 to $25\,\text{km}$ the BedMachine-CReSIS differences vary between 5 and $100\,\text{m}$, on the level of the uncertainties specified for BedMachine (Morlighem et al., 2017).

Below a distance of $16\,\text{km}$ only interpolation methods were used in BedMachine (see Appendix C). Independent CReSIS observations are in good agreement with each other (see cross-overs in, Fig. 13). Thus, we assume that CReSIS provides a





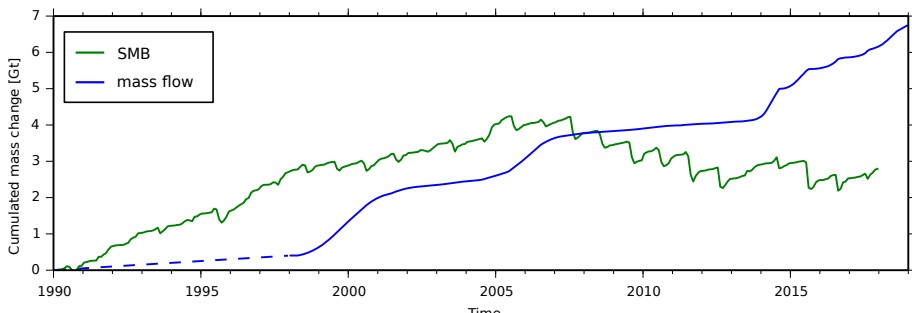

**Figure 12.** Cumulated mass flow through a cross-section of Harald Moltke Bræ close to the terminus (blue) and monthly cumulated SMB (green) summed over the drainage basin.

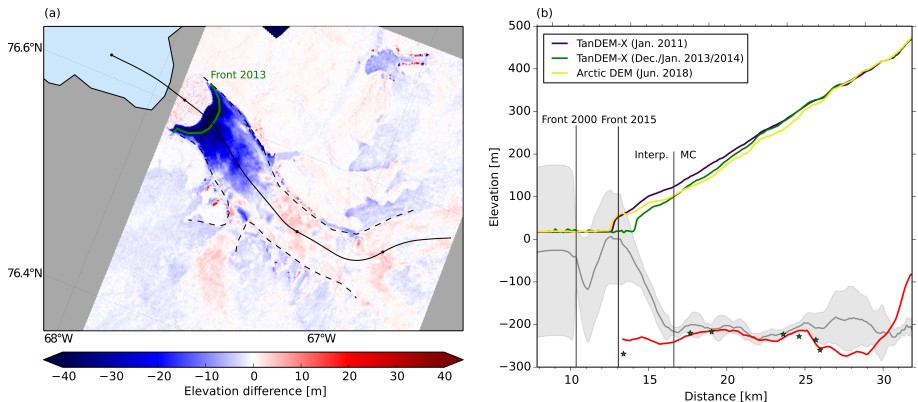

**Figure 13.** (a) Difference between ice surface elevations from two TanDEM-X observations (December/January 2013/2014 minus January 2011). (b) Profiles of bedrock topography from BedMachine (grey) and CReSIS (red) along the ground track of a flight path. Stated uncertainties for the BedMachine dataset (Morlighem et al., 2017) are indicated by the grey error band. Green stars mark the elevations from independent CReSIS measurements of August 2018 at the intersections of the ground tracks. Vertical lines mark the front positions of 2000 and 2015, and the boundary between the areas where BedMachine applied interpolation and a mass conservation approach (MC), respectively (Morlighem et al., 2017). Profiles of glacier surface height at different times are also shown. Subfigure a also shows the profile location (black line and black circles every 10 km along the profile) and the front position of Dec./Jan. 2013/14 (green line).

more accurate representation of the true bedrock topography close to the terminus and that the BedMachine sequence of a trough and a ridge at $10 - 12$ km and $13$ km, respectively, are interpolation artefact.

Surface elevations and elevation changes are also shown in Fig. 13. The difference between the 2013/2014 and the 2011 TDM
DEMs (Fig. 13a) reveals a significant ice thinning close to the glacier front by up to $30$ m in the 3 years preceding the 2013-2019 surge, whereas parts further up-glacier at $30$ km slightly thickened by an average of about $1 - 2$ m. As a consequence, the glacier surface was steepening (Fig. 13). Between 2013 and 2018, however, the glacier advanced and thickened in a small area at the terminus (Fig. 13) so that there the surface slope became gentler again. Thus, the geometry near the glacier front in 2018




was similar to that in 2011. At the same time, the ice surface height between 17 and 24 km decreased from winter 2013/2014

to 2018.

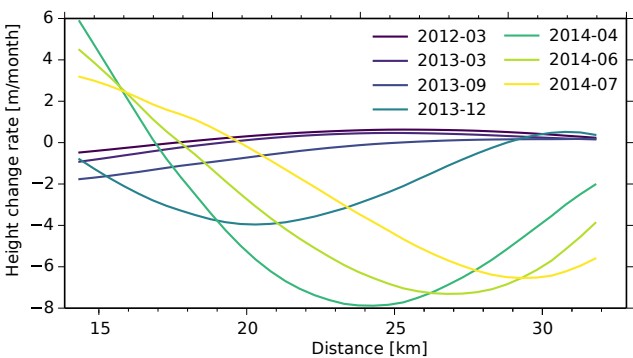

**Figure 14.** Dynamically caused ice-height changes derived from velocity profiles along a flow line (as marked in Fig. 5).

Monthly profiles of dynamic ice-height changes along a flow line are shown in Fig. 14. During the quiescent phase before the surge initiation in 2013, the spatial distribution of flow velocities induced a rather slow dynamic thinning of the glacier tongue between 14 and 20 km and a slow thickening further up-glacier. The pattern of thinning near the terminus is amplified to values up to 2 m/month in September 2013, shortly before the surge initiation. In December 2013 the extremum of surface lowering

has moved further upstream (to 20 km) and has reached a value of −4 m/month. In spring 2014, the massive acceleration of the entire lower part of Harald Moltke Bræ up to 25 km entails a rapid dynamic increase of surface height (3 to 6 m/month) in the lowest 5 km of the glacier and a rapid thinning (−8 to −6 m/month) further up-glacier. During spring 2014 this pattern appears to propagate upstream.

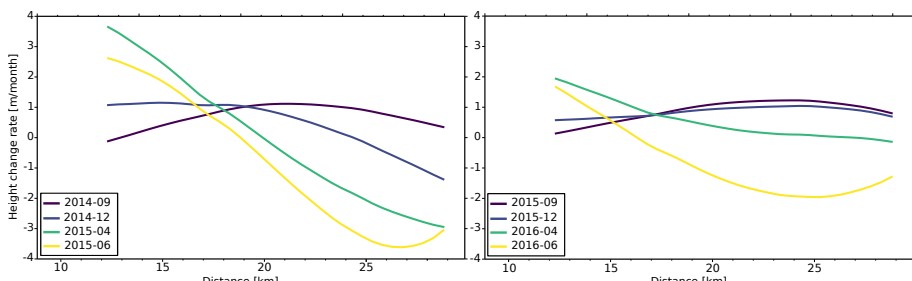

**Figure 15.** Dynamically caused ice-height changes derived from velocity profiles along a flow line for selected months during ice flow acceleration 2014/2015 (left) and 2015/2016 (right).

The accelerations of 2014/2015 and 2015/2016 (Fig. 15) differ from those of 2013/2014 as they were not preceded by a

rapid thinning. The dynamic height changes in 2014/2015 and 2015/2016 indicate overall a simultaneous acceleration of the glacier rather than an up-glacier propagation.



## 4 Interpretation and discussion

The flow velocity variations of Harald Moltke Bræ exhibit at least two different signals: Episodic surges and a seasonality with velocities abruptly decreasing in summer. The occurrence of a pronounced seasonality simultaneously to a surge can

be identified for the period 2013-2019. It would be conceivable that similar seasonal velocity changes were present during previous surges. In 1999, 2000 and 2005, any possible deceleration in July/August would have remained unobserved due to the lack of velocity data. In 2006 and 2007, flow velocities indeed decreased rapidly in summer similar to 2013-2019 (Fig. 4 and 6). Using the same data as the present study, yet at a different position, Rosenau (2014) showed that the velocities close to the terminus had decreased below $1\,\mathrm{m/day}$ before they increased again in 2007. The rapid acceleration in spring 2005 and the

velocity peaks in summer 2001, 2004 and 2007 are consistent with a seasonality that is characterized by maximum velocities in spring and early summer. Potentially, a seasonality of smaller amplitude was also present during the quiescent phases, but could not be identified due to the limited accuracy of the Landsat velocity fields. Thus, our main hypothesis for the explanation of the surge behaviour of Harald Moltke Bræ is as follows: The effect of seasonally changing external influences on ice dynamics is episodically amplified due to internal feedback mechanisms.

### 4.1 Types of seasonal velocity variations

We distinguish three different types of seasonality (a)-(c) at Harald Moltke Bræ: (a) In some years (e.g. 2016 and 2017), there were low and rather gently increasing velocities at the beginning of the year followed by a more pronounced acceleration from the onset of the melt season. Despite a significantly lower amplitude, the seasonality of 2001, 2004 and 2007 may also be consistent with type (a). (b) In 2014 and 2015, for instance, the velocities were already high at the beginning of the year (as

they have increased in autumn of the previous year), and remained high over several months in spring. In both cases (a) and (b), the flow velocities decrease rapidly in July or August. (c) During most of the quiescent phase, the velocities remained below $1\,\mathrm{m/day}$. A significant variation cannot be detected either because it does not occur or because of the limited accuracy of the Landsat data.

### 4.2 Seasonal mechanism

Types (a) and (b) correspond to the seasonality of the glacier types (2) and (3), respectively, identified by Moon et al. (2014) and Vijay et al. (2019). Harald Moltke Bræ switches between type-2 and type-3 behaviours. Further, during the surges, the seasonal variations being about $6\,\mathrm{m/day}$ clearly exceed the fluctuations of about $1-2\,\mathrm{m/day}$ observed by Moon et al. (2014).

As the deceleration in summer is correlated with the maximum of meltwater runoff, Moon et al. (2014) ascribed seasonal variations of the glacier types (2) and (3) to an annual switch between an efficient and an inefficient subglacial drainage system.

Similar to that, there may be a transition from an inefficient to an efficient subglacial drainage system in July/August in years with a seasonality of our type (a) or (b).

Some additional observations indicate a hydrological control on the seasonality at Harald Moltke Bræ: The formation of large meltwater lakes at the beginning of the summer when velocities are highest, the occurrence of a meltwater plume shortly





thereafter, the speedup at the onset of the melt season, and, further, the short-lived accelerations of ice flow right before the
slowdown in 2014 and 2015 as a presumed effect of enhanced basal water pressure which subsequently leads to the formation
of an effective drainage system.

According to Moon et al. (2014), type (3) differs from (2) in autumn as there is still enough meltwater available to raise the
water pressure significantly and, thus, to accelerate the ice flow. At Harald Moltke Bræ, we find a rather converse relationship:
Compared to years with low meltwater runoff (e.g. 2017), in years with larger meltwater runoff the ice flow slows down a bit
earlier (already in July instead of August), the velocities decrease to a lower level and the deceleration is not directly followed
by a rapid velocity increase in autumn (leading to type (b) in the following year) (see Fig. 8). Potentially, more meltwater leads
to a more efficient subglacial drainage system that prevails for a longer time in the year so that less meltwater is trapped at the
glacier base in autumn. Consequently, the comparably less pronounced decelerations in 2017 and 2018 may be interpreted as
an effect of lesser amounts of meltwater runoff.

**4.3    Is the surge 2013-2019 different from previous surges?**

There are several similarities between the surges 1999/2000 and 2005/2006: A duration of two years (years with velocities
exceeding $4\,\mathrm{m/day}$), persisting high velocities over at least 3 months in the second year of the surge (2000 and 2006) (see Fig.
5), and a year with slightly enhanced summer velocities following the surge. By contrast, the surge 2013-2019 had at least 6
years with velocities exceeding $4\,\mathrm{m/day}$. Further, the surge 2013-2019 was initiated by an abrupt acceleration in autumn/winter
reaching velocities of up to $10\,\mathrm{m/day}$, whereas the surge 2005/2006 began with an acceleration in spring 2005. It is likely that
all three observed surges, 1999/2000, 2005/2006 and 2013-2019, where modulated by a similar mechanisms of seasonality.

It must be considered that - before 2013 - the flow variability may be underestimated due to data gaps and larger time bases
(Appendix A) which involve large smoothing effects. Due to the retreat of the glacier front, the fixed reference point for the
velocities was closer to the glacier front in 2013 than at the times of previous surges. This is likely one reason why velocity
peaks in 2013 to 2019 are higher than during previous surges. Thus, the surges 1999/2000, 2005/2006 and 2013-2019 might
be more similar than suggested at a first glance in Fig. 1 and Fig. 6. The low velocities at the beginning of 2020 indicate a
transition to a new quiescent phase and, thus, a continuation of the surge behaviour.

As the observed accelerations 1926-1928 and 1954-1956 refer to the average over a period of 2 years, the true maximum
velocities are probably significantly higher than the documented velocities of 3.6 and $1\,\mathrm{m/day}$, respectively. In addition, the
exact timing of the surges is unknown. They may have lasted longer or shorter than 2 years. In summary, there is no argument
that the surge behaviour before 2000 was significantly different from that thereafter.

**4.4    Classification of flow patterns**

We distinguish between four different flow patterns (A-D) based on the spatial distribution of flow velocities. Each pattern is
associated with a certain profile of dynamic ice-height changes and, thus, has certain consequences for the stresses and the
mass redistribution in the glacier system.





(A) A first pattern is characterized by low velocities ($< 0.3\,\mathrm{m/day}$) in the lower part of the glacier and moderately higher velocities further upstream. This causes only minor dynamically induced changes in the receiving area and a dynamic thickening in the reservoir area. This pattern can be found for much of the quiescent phases, e.g. in March 2012 (see Fig. 10). (B) The second pattern equals (A), except for an area near the glacier front where the velocities exceed $1\,\mathrm{m/day}$. This pattern

arises shortly before the surge initiation in September 2013 (see Fig. 7). Thus, additionally to the thickening in the reservoir area a pronounced thinning occurs directly at the glacier front. (C) A third pattern has a steeply sloped velocity profile (with a maximum at the glacier front) in the lower $10\,\mathrm{km}$ of the glacier, whereas larger parts further upstream remain at low velocities similar to patterns (A) and (B). This was the case in December 2013 (see Fig. 7), for instance. This pattern may be associated with only moderate height changes at the glacier front and in the upper parts of the glacier, whereas the middle part of the

glacier is rapidly thinning (see Fig. 14). This results in a decrease of surface slope at the terminus and a significant increase of surface slope in the middle part. (D) The fourth pattern mostly reverses pattern (A) so that the glacier thickens in the receiving area and thins in the reservoir area (see Fig. 14). Thus, the effect of the ice dynamics on the glacier geometry during the quiescent phase is compensated. This pattern is shown, e.g., by the velocities in spring and early summer 2014 (see Fig. 7).

During the initiation of the surge 2013-2014, these flow patterns followed each other in the sequence (A)-(B)-(C)-(D).

Similar to winter 2013/2014, the velocity profiles in 1999 and 2005 (first year of the surge) were also steeper than in the following years 2000 and 2006 (second year of the surge). Hence, there might have been a similar sequence (A)-(B)-(C)-(D) initiating all surges. This sequence reflects an evolution of dynamic changes within the glacier where first, through pattern (B), mass from the middle part of the glacier is mobilized to compensate the mass deficit at the glacier front leading to (C). Subsequently, the surplus of mass in the upper part of the glacier is set into motion to compensate the mass deficit in both the

middle and the lower part resulting in pattern (D).

In the years after the surge initiation, the sequence appears to be different. There is a rather direct switch between the (A) and (D), whereas the patterns (B) and (C) are largely absent (see Fig. 15).

### 4.5 Up-glacier propagation of the surges

The sequence of patterns (A)-(B)-(C)-(D) is consistent with an up-glacier propagation of the surge. In the consecutive years,

however, the direct switch between (A) and (D) reflects a rather uniform acceleration of most of the ablation zone of Harald Moltke Bræ. Based on these findings, we conclude that once the entire glacier has accelerated, large parts of the glacier are set into a condition to accelerate again in the following years.

Surge looped moraines typically indicate a down-glacier surge propagation. At Harald Moltke Bræ, however, no such looped moraines were observed at the confluence of Blue Ice Valley Glacier and Harald Moltke Bræ. This is a further indicator for an

up-glacier propagation of the surge.

### 4.6 Glacier geometry during the quiescent phases

Similar to other surge-type glaciers, the quiescent phase of Harald Moltke Bræ is characterized by slower flow in the lower part und faster flow in the upper part. However, the transition between slower and faster flow is rather smooth which implies



just a slight dynamic thickening (few metres over three years, see Fig. 13). We therefore hypothesize that for the initiation
of the surge, the increase of driving stresses in this transition zone is less important than the decrease of resisting stress
at the glacier front implied by the frontal retreat. The findings are opposite down-glacier propagating surges e.g. observed at
Variegated Glacier (Raymond, 1987) and at Bivachny Glacier (Wendt et al., 2017). The pronounced thinning at the glacier front
is compatible with an up-glacier propagating surge observed at Aventsmarksbrae (Sevestre et al., 2018). Thus, a significantly
larger thinning at the glacier front compared to a lesser thickening in the upper part of the glacier could be a key factor for the
surge initiation at the terminus and an up-glacier propagation.

### 4.7  Potential causes and triggers of the surges

On long time scales, the dynamics of Harald Moltke Bræ may be determined by externally driven influences such as SMB and
the long-term retreat of the glacier front. On the interannual time scale, however, the flow velocity does not react simultaneously
to external drivers such as the terminus retreat and the cumulated SMB. Instead, local conditions might restrict the ice flow
during quiescent phase and facilitate an increasing instability that finally results in a surge.

    Several factors may potentially restrict the ice flow in the lower part of the glacier during a quiescent phase: A cold glacier
base that is partly frozen to the glacier bed (corresponding to the thermally driven mechanism, maybe related to strengthened
subglacial till), a subglacial drainage system remaining efficient over several years (corresponding to the hydrologically driven
mechanism), a bump in the bedrock topography, a dynamic interaction of Blue Ice Valley Glacier and Harald Moltke Bræ, and
the absence, or closure of crevasses preventing meltwater from reaching the glacier base.

    The location of Harald Moltke Bræ would be favourable for a thermally driven surge. Rather low temperatures prevailing
in northern Greenland may provide polythermal conditions so that the glacier could be partly frozen to its bed. Further, the
marine-terminating glacier might have deformable sediments at its base which could facilitate such a thermally driven feedback.
However, the rapid initiation of the surge in 2013 is unlike the typical gradual initiation of thermally driven surges. Moreover,
intermediate decelerations during summer, when the glacier goes back to a state similar to the quiescent phase, cannot be
explained by an abrupt freezing of the glacier base or a strengthening of subglacial sediments. Hydrologically driven surges,
on the other hand, are often characterized by a downward propagation of the surge associated with the downward propagation
of the trapped water beneath the glacier (Wendt et al., 2017; Raymond, 1987). This could not be observed at Harald Moltke
Bræ. However, the presence of a seasonality parallel to surges has been taken as a clear indicator for a hydrologically driven
mechanism at other glaciers (Wendt et al., 2017; Raymond, 1987). Mayer et al. (2011) explained the seasonal modulation of a
surge by the Kamb drainage-switching theory. We conclude that the hydrological mechanism provides a overall more plausible
explanation for the surges at Harald Moltke Bræ than a thermally driven mechanism.

    Wendt et al. (2017) discussed a bump in the glacier bed as cause of restricted ice flow during the quiescent phase of Bivachny
Glacier. At Harald Moltke Bræ, there is a bump at about 25 km according to the CReSIS dataset, which is, however, small
compared to the ice thickness. Closer to the terminus, the large uncertainties of the BedMachine dataset does not allow to
identify relevant features of the bedrock topography.





The surges of Harald Moltke Bræ may develop as follows: During the quiescent phase, the combined effect of low flow velocities and a negative mass balance in the ablation zone involve a thinning and steepening of the glacier tongue and a retreat of the glacier front. Observations from CReSIS indicate a rather thick marine-terminating glacier front of more than 200 m
which could facilitate the thinning and the retreat. These factors may lead to a decrease of resisting forces at the terminus. Additionally, at a thick glacier, the steepening of glacier due to thinning at the terminus could increase the driving stress (Sevestre et al., 2018). The resulting large net stress in flow direction may cause a small area at the glacier front to accelerate (flow pattern B). This induces different effects that facilitate a further acceleration and the upward propagation of the surge: crevassing of the ice surface enabling more meltwater to reach the glacier base (Sevestre et al., 2018), weakening of subglacial
till (deformation of the glacier bed), longitudinal stresses (tension) and a further surface thinning/steepening. As a result, the glacier may change into pattern (C), which, in turn, transfers these processes further up-glacier leading to pattern (D). Once the entire ablation zone of Harald Moltke Bræ was affected by the sequence (A)-(B)-(C)-(D), the externally driven seasonal changes of meltwater availability could have a more pronounced effect as crevasses have spread over large parts of the glacier and subglacial till has weakened.

This could explain that in years after the year of surge initiation there is a simultaneous acceleration of large parts of the glacier according to a direct flipping between (A) and (D) rather than an up-glacier propagation. Pattern (D) causes the ice surface to become more gently sloped. Such a decrease of surface slope could stabilize the glacier during the surge and, thus, lead to a reverse feedback: A decreasing annual amplitude causing a gradual closure of the crevasses and, possibly, a strengthening of the subglacial sediments until the glacier enters a quiescent phase. Such an evolution is consistent with the
velocity peaks in summer 2001 and 2007 following the surges.

Both the SMB and the retreat of the glacier front could determine the time of initiation and the length of the surge cycle. The 2013-2019 surge and its preceding quiescent phase lasted longer (5 years and 6 years) than the surge and the quiescent phase in the preceding cycle (5 and 2 years, respectively). This could be related to the SMB in the drainage basin being negative after 2006. Also, the intermediate annual retreat of the glacier front in 2016 and 2017 could have prevented the glacier from
stabilizing such that in 2018 and 2019 velocities of more than 4 m/day are still reached.

## 5    Conclusions

By combining four different remotely sensed velocity data sets we estimated a monthly velocity time series for Harald Moltke Bræ with high spatial and temporal resolution. Based on this time series we identified mainly two different signals of velocity variations close to the terminus of Harald Moltke Bræ: episodic surges and a pronounced seasonality. As we assume that there
is a similar seasonality in most years of the observation period, we interpret the surges as phases with a strongly amplified seasonal amplitude.

The annual flow velocities of Harald Moltke Bræ were high over a relatively long period of 6 years from 2013 to 2019, which has not been observed in this way before. However, flow velocities remaining low in autumn/winter 2019 and spring 2020 indicate the beginning of a new quiescent phase. Thus, the high velocities between 2013 and 2019 may constitute a longer

surge than previous ones, but no fundamental change in the flow regime. We therefore assume that Harald Moltke Bræ is likely
        to maintain its surge behaviour.

        Regarding temporal velocity variability, we identified types of seasonality which indicate a hydrological control of the
        seasonal velocity changes. The time of a rapid deceleration in July or August suggests a switch between an inefficient and an
        efficient subglacial drainage system due to the changing amount of meltwater availability. This, however, does not provide an
explanation for the significant increase of seasonal amplitude during the surges.

        Over the past two decades, the overall discharge of the glacier was larger than the overall ice mass accumulation. Thus, the
        overall thickness of the glacier was decreasing which could be a reaction to the observed long-term retreat of Harald Moltke
        Bræ. However, the glacier does not react instantaneously to the terminus retreat by dynamic thinning. The reason might be a
        restricted ice flow during the quiescent phases due to internal factors, probably related to the glacial hydrology. This possibly
causes an increasing instability and results in an alternation of two reverse feedback mechanisms which involve pronounced
        acceleration and deceleration, respectively.

        By distinguishing between different patterns of spatial distribution of flow velocities we could demonstrate that the surge
        develops first at the glacier front, and that it is propagating rapidly up-glacier within a few months thereafter. The seasonal
        amplitude remains high in the years after the year of surge initiation with a rather simultaneous acceleration of the entire
ablation zone of Harald Moltke Bræ. Thus, we assume that during the surge there are favourable conditions for an enhanced
        effect of the seasonally changing meltwater availability on the ice flow. This could be a crevassed glacier surface and possibly
        weakened subglacial till. As the seasonal amplitude decreases gradually after the surges, there might be a stabilization related
        to the flattening of the glacier with increasing duration of the surge.

        The presence of additional factors that could restrict ice flow during quiescent phase such as a bump in the bedrock topogra-
phy or a reversely sloped bed at the terminus could not be identified. However, an adjacent lake at the northern side of Harald
        Moltke Bræ and the confluence between the Blue Ice Valley Glacier and Harald Moltke Bræ could be factors that facilitate the
        surge behaviour.

        External factors may not be the cause for the surge behaviour but could play an important role for the time of the initiation
        and the duration of the surges. Particularly, the rapid retreat of the glacier front with an acceleration and a thinning of a small
area at the terminus during quiescent phase may determine the way in which the surges develop. The marine-terminating glacier
        front could, thus, be an important factor for the surge behaviour of Harald Moltke Bræ.

## Appendix A: Velocity data sets and determination of joint velocity time series

### Initial velocity fields and pre-refinements

The velocities derived from remote sensing techniques represent average values over the given time basis, the time basis
        being the time difference between the acquisition epochs of the two images used for the velocity estimation. Thus, long time
        bases might result in a stronger temporal smoothing. Regarding the quality and the temporal coverage of the Landsat velocity



fields, the deployment of the Operational Land Imager (OLI) at Landsat 8 launched in 2013 was a significant improvement. It extended the yearly image acquisition to the period March-November, while the acquisition period was May-October before

Landsat-8. It also improved the spatial resolution from 300 to 150 m. These improvements, in turn, enabled to use image pairs with shorter time bases and, thus, the resolution of rather short-term velocity fluctuations. The initial Landsat data set contains velocity fields with very different time bases (Tab. 1) which leads to varying smoothing effects (Fig. A1).

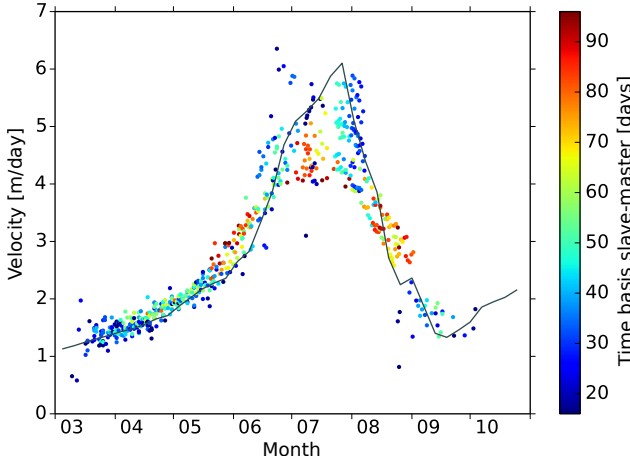

**Figure A1.** Landsat velocities at the terminus of Harald Moltke Bræ in the year 2017, color scaled with the time basis. For comparison, the velocities from Sentinel-1 (P) are plotted as grey line.

Therefore, we reject those velocity fields with a time basis longer than 60 days in order to avoid strong smoothing effects. Further, we do not use Landsat ice flow velocity fields with a time basis shorter than 25 days as these are assumed to be of

comparatively lower accuracy.

Velocities are analysed in form of spatial fields, time series with respect to a point close to the glacier front or velocity profiles along a flow line.

**Estimation of monthly velocity fields**

To ensure consistency when determining a joint time series, all given data grids are transformed to the coordinate system of the Landsat data (Polar Stereographic Grid North). Subsequently, monthly medians are calculated separately for each data set. The resulting monthly time series are merged by computing the monthly mean for each cell. A similar approach was applied

to determine semimonthly velocity fields.





**Accuracy of the velocity fields**

The point velocities derived from different methods (Fig. 1) agree well, with only minor deviations being generally less than
$0.5\,\mathrm{m/day}$. The uncertainties specified for Sentinel-1 (P) and TerraSAR-X ($0.1$ and $0.01\,\mathrm{m/day}$, respectively), are significantly
lower than the uncertainties for Landsat ($0.3\,\mathrm{m/day}$, Tab. A1). However, methods applied for uncertainty assessment may differ
between the datasets. A direct comparison of the monthly medians shows that the mean deviations between Landsat, Sentinel-
1(P) and TerraSAR in the ablation zone of Harald Moltke Bræ range from $0.1$ to $0.4\,\mathrm{m/day}$ in phases with low velocities
($<2\,\mathrm{m/day}$), and from $0.3$ to $1.3\,\mathrm{m/day}$ in phases with high velocities ($>5\,\mathrm{m/day}$). We conclude that the accuracy of the
estimated joint time series of monthly velocities significantly dependends on the magnitude and the temporal variability of the
flow velocities.

**Table A1.** First row: uncertainties specified for each data set. Row 2-4: Root-Mean-Square (RMS) differences between the two data sets
indicated in the line and column heads, evaluated within the ablation area of Harald Moltke Bræ. As the temporal variability of the flow
velocities within one month is not negligible, we distinguish between months with slow maximum velocities ($<2\,\mathrm{m/day}$) and high maximum
velocities ($>5\,\mathrm{m/day}$, parenthesis). Unit are $\mathrm{m/day}$.

|  | Landsat | Sentinel (DTU) | Sentinel (PROMICE) | TerraSAR-X |
| --- | --- | --- | --- | --- |
| given uncertainty | 0.3 | - | 0.1 | 0.01 |
| Landsat |  | 1 (1.5) | 0.1 (0.3) | 0.1 (0.4) |
| Sentinel (DTU) |  |  | 0.4 (0.7) | 0.3 (1.3) |
| Sentinel (PROMICE) |  |  |  | 0.2 (-) |

**Appendix B:  Ice front positions**

To compute the mean ice front position change we define a rectangle approximately having the width of the glacier as shown
in Fig. B1 (Moon et al., 2008). The glacial area enclosed by the front positions and the rectangle is determined and divided by
the width of the rectangle to obtain the average change of the front position.

**Appendix C:  Bedrock geometry**

As shown in Fig. 13, the BedMachine bedrock topography exhibits a significant depression close to the front position of the year
2000 (terminus position used in BedMachine), a pronounced elevation at the terminus position of 2015 and a sharply down-
sloping bedrock further upstream. The methods applied for bedrock topography determination depend on surface properties
and data availability (Morlighem et al., 2017). The mentioned distinct topographic features coincide with transitions between
different methods and different associated uncertainty levels (Fig. C1a,c). In particular, different interpolation methods prevail





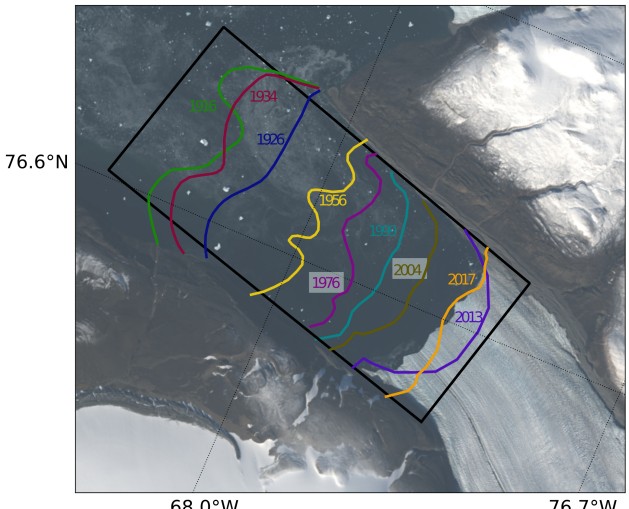

**Figure B1.** Digitized ice front positions and rectangle used for the estimation of the mean front position change. Background image: Landsat 8 scene of 2014 (USGS, 2019).

near the glacier front. A synthetic approach was applied for Wolstenholme Fjord (Williams et al., 2017), where only little bathymetry data were available. There, the assessed uncertainties are as large as $250\,\mathrm{m}$ (Morlighem et al., 2017).

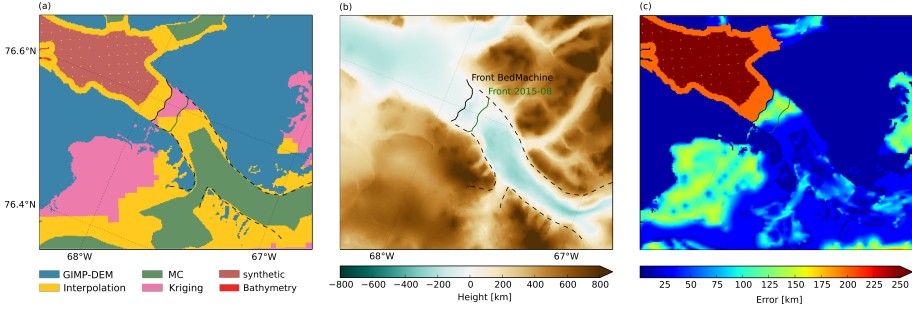

**Figure C1.** Comparison of the applied methods, the resulting bedrock topography and the corresponding error estimations by Morlighem et al. (2017). Elevations are given as ellipsoidal heights with respect to the WGS84 ellipsoid.

## Appendix D: Dynamic ice-height change

To determine the dynamic ice-height change using Eq. 1, profiles of ice thickness and velocity are used. The ice thickness for the dynamic height change estimation is approximated by the difference between the ice surface elevation from the ArcticDEM and bedrock topography. Test results for the dynamic height changes have been computed with the bedrock topography from CReSIS and from BedMachnine for distances $> 16\,\mathrm{km}$ along the profile. The results showed approximately the same overall





pattern of dynamic height changes as when using a constant height of $-250\,\text{m}$. The steeply sloping profile from BedMachine is

assumed to be an artefact due to interpolation methods. Minor structures ($< 20\,\text{m}$) in the bedrock topography can be neglected

for their small effect on the derived patterns of dynamic ice-height changes. Thus, we used a constant bedrock topography of

$-250\,\text{m}$ to compute rough patterns of dynamic height changes.



## Appendix E: Visually inspected features in Landsat images

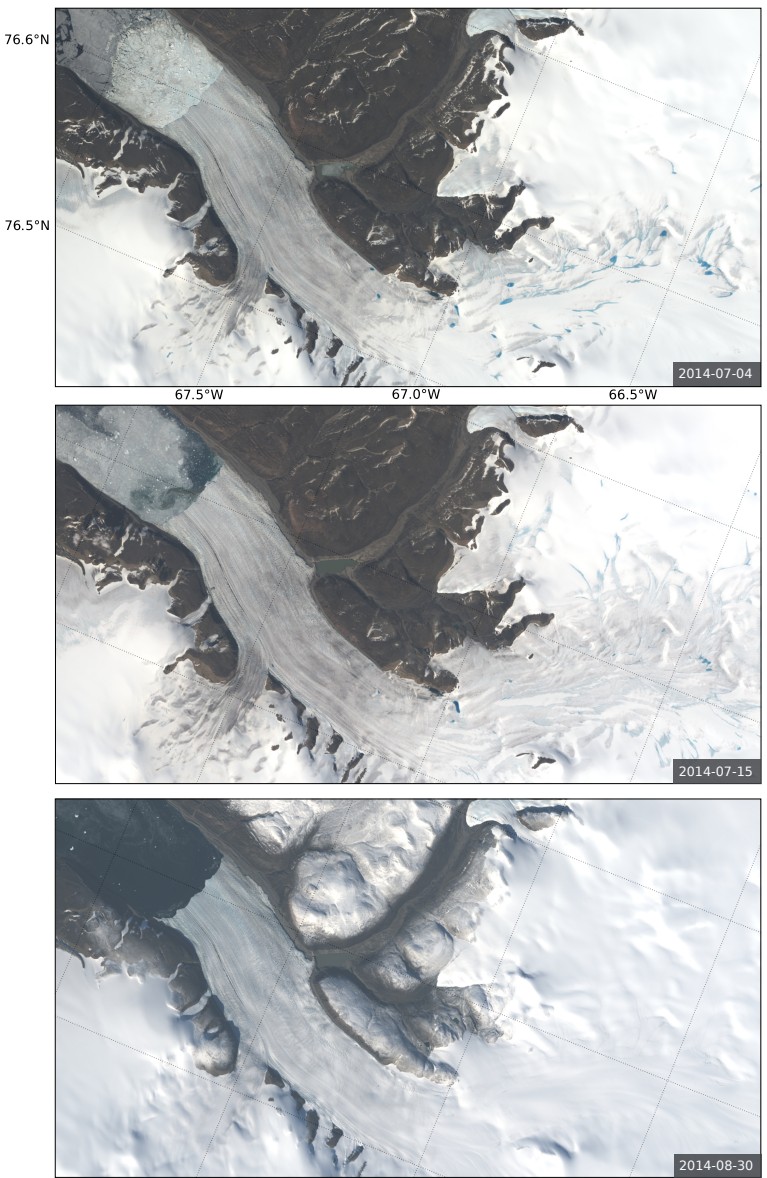

**Figure E1.** Exemplary three different Landsat scenes during the phase of rapid deceleration in early summer 2014 (USGS, 2019).



*Author contributions.* LM conducted the study with the support of MH and MS. SV provided the Sentinel-1 (D) velocity data. BE and RR contributed the Landsat velocity data. LK and DF processed the TDM DEMs. The manuscript was prepared by LM and revised by all other authors.

*Competing interests.* The authors declare that they have no conflict of interest.





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
