# Peer review of "Surges of Harald Moltke Bræ, north-west Greenland: Seasonal modulation and initiation at the terminus"

_The Cryosphere, 2020_

## Referee Comment (RC1) · Vanessa Round (Referee) · 13 Dec 2020

In this paper the authors present a characterisation and analysis of the dynamics of Harald Moltke Brae, a marine terminating glacier in north-west Greenland which experiences multi-year periods of accelerated glacier flow (surging). They use a variety of optical and radar remote sensing products from 1998 onwards to deduct surface velocities and surface elevation models using well established and cited methods, as well as to observe ice front positions and hydrological features. Several patterns of seasonal velocity variation and spatial flow distribution are identified and used in describing and interpreting the observed behaviour.

[Figure]

The strong seasonal flow modulation observed during the most recent surge is consistent with possible hydrological controls which have been discussed in existing literature (i.e. acceleration with meltwater onset and rapid deceleration with the switch from an inefficient to an efficient subglacial drainage system at around meltwater peak). Surges are reported to initiate at the terminus and ice flow acceleration propagates up-glacier, and possible mechanisms for this are discussed. The analysis of whether the most recent surge was significantly different to past surges is somewhat unclear and could be expanded on. It will naturally be very interesting to see how the situation evolves into the future.

This paper is well written and will surely be of interest to readers of the Cryosphere, in particular those interested in the phenomenon of glacier surging. It presents a thorough and detailed picture of the dynamical behaviour of Harald Moltke Brae since 1998, bringing together a lot of observational data and contributing an interesting discussion. I have a number of specific comments and questions which follow below.

SPECIFIC COMMENTS:

L 6-7: From the results presented it doesn't seem like there is enough evidence to state that there is similar seasonality during the quiescent phases. As you say in L261, there could potentially be seasonality present during the quiescent phases, but it could not be identified due to the limited accuracy of the Landsat velocity fields.

L 7: The choice of the word 'peculiar' suggests to me that the seasonal amplitude during surging is something observed only at Harald Moltke Brae, which is not the case – perhaps 'significant', 'noteworthy', 'interesting' or something along those lines would avoid this potential misunderstanding.

L 26-27 and Figure 1: It is stated that velocity remained low from the end of 2019 to the beginning of 2020 but from Figure 1 it looks like the low velocities extend to mid/late 2020 although it is hard to see for sure when the last data point is. Could the text be updated to state when (which month) in 2020 the data extends until? It would be

interesting to know whether the velocity remained low over the summer 2020.

L 89: Is it possible that the Sentinel and TerraSAR-X data can be used to get velocity fields with temporal resolution finer than monthly, given that the time basis and temporal resolution is less than a month? If this is the case, the use of a monthly averaged dataset could mean temporal resolution is being lost. I think that the use of a monthly combined velocity dataset is reasonable, but have you analysed the individual datasets to make sure you are not losing potentially valuable information on a finer temporal scale, particularly around the rapid summer decelerations? When looking at the surge of Kyagar Glacier (doi:10.5194/tc-11-723-2017) we used TerraSAR-X data to compare velocities over consecutive 11 day periods to identify similar rapid summer decelerations.

Table 1: Could you provide a specific range of days for the Landsat temporal resolution rather than 'few days'?

L 99: It states that 4 DEMs are used but only three seem to be referred to – the ActicDEM and two interferometric DEMs from January 2011 and Dec/Jan 2013/2014. Is there something missing, or is the fourth DEM referring to the computed ice-height change rates?

Figure 3: The shading representing the most recent surge seems to cover less than the six years of reported surging. Is there a reason for this or is it an oversight? Additionally, I think it might be clearer to remove the black line joining the observed front position points, to better represent the discontinuous nature of the historical dataset.

Figure 6: Is there a reason not to include 2019 or even 2020 in this figure considering that the surge continued through 2019? It would give a more complete picture of this very recent surge if the data extended as far as possible, including termination of the surge in late 2019 or in 2020. I also recommend repeating the description of parameters shown on panels rather than refer back to Figure 4 – this allows Figure 6 to be understood in isolation.

Figure 7: The colours in the flow velocity profile plots are difficult to distinguish, especially 2013-09 and 2013-12. Please consider using a range with more contrast in these types of figures (e.g. also Figure 14, 15) or perhaps even different line patterns.

Figure 8: The year 2018 is almost impossible to detect in the bottom panel. Could you find a way to make it visible or if it is hidden behind one of the other years perhaps make a note of this in the caption.

L 209: I assume this is referring to a lack of significant year-to-year changes in the lake, however it could be useful to provide a brief description of the seasonal patterns observed (lake formation in summer). In section 4.2 it is noted that large meltwater lakes form at the beginning of summer so it would be useful to make note of that here in the results section.

Figure 11: I suggest considering switching the axes in this figure i.e. stacking the years vertically, to make it easier to compare the timing of events between the years (as described from L 206 onwards).

Figure 14: It would be helpful to show a horizontal line at 0 to make it easier to distinguish between positive/negative height gain (especially for the earlier dates for where the difference is more subtle). The same applies to Figure 15.

L 275: It would be nice to briefly summarise the glacier types 2 and 3 identified by Moon et al. here.

L 282: The observation of seasonal meltwater lakes was not noted in the results and should perhaps be included in section 3.5 too.

L 307: Regarding the low velocities at the beginning of 2020 – when is the latest available velocity data, i.e. from which month in 2020 and in particular do they extend into the spring/summer?

L 310: The summary sentence here seems a bit subjective and also contradicts the statement in the abstract that the most recent surge 'lasted significantly longer' than
previously observed surges or the passage from L 74-76. If not in terms of maximum velocity then at least in terms of surge duration the most recent surge does seem (in my subjective opinion) significantly different to at least the two well observed surges before it. The various arguments presented in section 4.3 mostly relate to limitations in the data (e.g. lack of data pre 2013, greater smoothing on velocity maxima in the earlier Landsat data). So in summary, it would be more correct to say there is insufficient evidence (historical data) to conclude whether the surge behaviour has changed since 2000.

L 319: The example of September 2013 doesn't show the moderately higher velocities further upstream which is described for pattern A, but rather seems to show velocity decreases up the glacier. Is it just the high velocity at the terminus which is the defining feature of pattern B?

L 326: I like this concept of categorising the various flow profles but the difference between C and D isn't very clearly defined – both of these patterns show high velocities with the maximum at the glacier front. Is the difference that with C the velocities high up the glacier are lower and hence the overall profile steeper, or that velocities are higher overall for D? Also it might be helpful to add that while pattern D reverses A, the timescales will be quite different because of the difference in the magnitude of velocity between these two patterns.

TECHNICAL CORRECTIONS:

L 227: There is a typo on the stated depth of 230 km (should be m). L 251: Typo 'glaclier' rather than glacier.

---

## Referee Comment (RC2) · Anonymous Referee #2 · 18 Jan 2021

This paper provides an interesting review of the surges of Harald Moltke Brae in NW Greenland, and their unique characteristics. In particular, the remote sensing data demonstrates that the glacier undergoes dramatic seasonal variability in motion during a surge (by up to an order of magnitude), and that the most recent surge lasted for 6 years, compared to a duration of 2 years for the two surges prior to it. The surges also initiate at the glacier terminus and propagate upglacier. Such strong seasonal variability in motion during a surge hasn't been reported for any glacier before, and there aren't many other reports of upglacier propagating surges. The results from this paper can therefore help advance general glaciological understanding of glacier surges, and the wide variability in their form and extent.

[Figure]

The data is generally well presented, with lots of illustrative figures, although there are several places where some clarification of the methods and results, and improvements to the figures, would help. These are detailed below, together with other technical corrections. There are also several places where better referencing to existing literature would be useful. I find that the arguments in the Discussion and Conclusions concerning the causes of surges and their connections to basal shear stress and changes in the subglacial hydrological system are currently a bit weak, and would really be strengthened by modelling, but I expect that this modelling is beyond the scope of this paper.

L5: I would suggest saying 'annual velocity' or 'seasonal velocity' here to make it clear that this velocity variability occurred on an annual, repeating basis during the 6 year surge, rather than just once during it.

L8: I think that the wording is better as 'constitute a glacier surge', since you're talking about a singular surge each time

L10: 'involving' would be better worded as 'examining'

L19-L41: I find the Introduction quite strange as it opens with a presentation of the results and study area, before providing any of the background or methods that would usually be expected in a paper. I therefore suggest moving the text from L19-L27 and Figure 1 to the Results, and the text from L28-L41 to a new section called 'Study Area' after the Introduction. The Introduction would then start on L42, although this first sentence might need to be modified.

L35: I would delete 'As another remarkable feature,', and start this sentence with 'A 3 km long and l km wide lake. . .', as I'm unclear as to how this lake is particularly remarkable in this part of the paper

L44-45: more recent papers and reviews suggest that the length of the active and quiescent phases of surge-type glaciers can be longer than what you state. For exnone

ample, Jiskoot (2015) states that the quiescent phase lasts for 10s to 100s of years, while the active phase lasts for 10-15 years. See: Jiskoot H. (2011) Glacier Surging. In: Singh V.P., Singh P., Haritashya U.K. (eds) Encyclopedia of Snow, Ice and Glaciers. Encyclopedia of Earth Sciences Series, pp 415-428. Springer, Dordrecht. https://doi.org/10.1007/978-90-481-2642-2_559.

L51: The study of Monacobreen by Murray et al. (2003) is one of the first to have reported an upglacier propagating surge, so I think that it should be referenced here

L56: you should also mention the large surge cluster in east Greenland, reported by studies such as Sevestre and Benn (2015) and others

L61: You need some words to introduce this sentence, such as: 'In mechanism (A), the base of a polythermal glacier...'. Similar to introduce (B) on L63

L78: I would delete 'extraordinary' as several other studies have previously documented surges initiated at the glacier terminus (e.g., see comment for L51)

L85: I don't understand what 'suited' refers to here. Perhaps you mean 'suitable', but in that case you need to describe why the images would be suitable. In this section also make it clear that the black triangle in Fig. 2 is the point that all velocity time series were derived for – it took me a long time to spot this information in the figure caption.

L89: would be useful to provide some numbers here to define what you mean by 'spatial and temporal coverage as high as possible'. e.g., max resolution, max temporal coverage Table 1 caption: change 'Overview over...' to 'Overview of...'

L99: You state that you use four different DEMs, but only list three

L112: it would be useful to provide a reference or two for the choice of 0.9 to convert the surface velocity to depth-average velocity. For example, Cuffey and Paterson (2010) provide a discussion of this: Cuffey, K.M. and Paterson, W.S.B., 2010. The Physics of Glaciers. Academic Press

[Figure]

L119-122: provide the resolution of your basal topography data

L125: in the caption for Fig. 4 it's stated that the SMB is calculated from the difference between precipitation and runoff (i.e., it's not calculated independently), so that should be made clear here. This also seems to implicitly assume that no mass is lost from calving at the terminus, but from Fig. 5 and 7c it looks as if this could be important. Can you therefore address whether this is accounted for, and what implications this has for your SMB data? This is partly discussed in Section 3.6, but it's unclear whether your SMB values are adjusted for the calving flux.

L127: would be useful if you can show the location of this cross section on Fig. 2

L134: This seems to be the Results section, so it would be good to include that in the title

L136/7: would be better expressed as '. . .advance of the terminus which interrupted its long-term retreat'

L152: change 'exceptional' to 'exceptionally'. Can you also talk about whether this high precipitation fell as rain or snow, and what time during the year it fell? For example, high rainfall in the summer might have a different impact on dynamics compared to high snowfall in the winter.

L172: change to 'compensate for the calving rate'

Figures 4 & 6: please use letters (a), (b) and (c) to label and refer to the different figure parts, consistent with the other figures. In the figure caption, the text '(Black triangle in Fig. 1)' should refer to Fig. 2. It would also be useful to see the surge periods labelled on these figures (e.g., by lightly shading the background), similar to what you show in Fig. 3, so that it's clear as to how their stated start and end times of the surges match up with the velocities and frontal positions.

Figure 8a seems to show the velocities at quite a bit higher temporal resolution than in Fig. 6. The better temporal resolution allows the variations to be more clearly seen,

[Figure]

particularly for years such as 2014 and 2015, so can you plot the full high resolution dataset (and associated errors) in Fig. 6?

L214-220: this discussion only refers to mass lost by flow through the cross-section close to the glacier front. However, mass is also lost by retreat of the glacier terminus (and mass gained when it advances), so this should also be accounted for when presenting the mass numbers and balancing them against the flux through the cross-section and the SMB.

Figure 11: there is insufficient information provided in the figure caption or methods to understand what exactly was measured for these visual features, and how a distinction is made between 'moderate' and 'strong' conditions. The methods refer to Appendix E, but this simply consists of three satellite images without any description. Please provide more information so that the reader can understand what was done, and what was measured.

L227: change '230 km' to '230 m'

L251: change 'glaclier' to 'glacier'

L252: at the start of this section I would like to see a few sentences to describe exactly how you defined a surge, including its start and end date. For example, is this just based on velocity variations, or also on things such as changes in terminus position, calving rate, surface crevassing, etc.? This is needed to put the remainder of this section in perspective, particularly because the 2013-19 surge was so much longer than the previous two surges.

L266-273: here and elsewhere in this section it would be useful to make better reference to the figures

L283: please provide some more detail about the large meltwater lakes. E.g., are these supraglacial? Ice marginal? Where on the glacier do they form?

L286: I think that you mean 'efficient drainage system' here, rather than 'effective'?

[Figure]

L301: change to 'were modulated by a similar mechanism'

L322: can you provide the maximum velocity for the third pattern?

L316-337: I find it a bit hard to follow and visualize the flow patterns based on the text descriptions here. To make them easier to understand, could you add a schematic or cartoon that shows the velocities and surface elevation changes associated with each pattern (A, B, C, D), and the sequence that occurs during surge initiation (A-B-C-D) and after surge initiation (A-D)? This will also help with the description of upglacier propagation vs uniform acceleration described in Section 4.5.

L343: can you provide a reference to support the statement that 'Surge looped moraines typically indicate a down-glacier surge propagation'?

L351: change to 'opposite to down-glacier'

L353: would also be useful to make comparisons with other up-glacier propagating surges, as well as the findings of this paper (particularly for Section 4.7): Thøgersen, K. et al. 2019. Rate-and-state friction explains glacier surge propagation. Nature Communications, 10, 2823.

L366: can you provide more information about this? E.g., what are the mean annual temperatures here? Have there been any direct measurements or evidence of poly-thermal conditions at this glacier or other nearby ones of a similar size?

L375: please describe what the 'Kamb drainage-switching theory' is for any readers who might not be familiar with it

L389-394: do you have any observations of crevasse formation at Harald Moltke Brae from the remote sensing imagery that provide support for the statements made here? It could be useful to add them as a visual feature to Fig. 11.

L421: it would be useful to provide some numbers here for the mass imbalance, including the mass due to terminus retreat (related to comment for L214-220)

L431: was any change in crevasse formation actually observed (similar comment to L389-394)? It could also be useful to refer to the visual indicators from Fig. 11 here or elsewhere in the Conclusions to help back up your statements.

L450: I don't understand the reference to improvement in spatial resolution here, when both Landsat 7 and 8 have a multispectral resolution of 30 m and panchromatic resolution of 15 m

L450: change to 'enabled use of image pairs...'

Figure A1: there have been calls to stop using the terms 'slave-master' in remote sensing as they can be interpreted as colonial terms. Something such as 'reference' and 'secondary' is better. See, for example: https://earthenable.wordpress.com/2020/08/11/new-insar-terminology-coming-in-vogue-master-slave-to-reference-secondary/

Appendix E: text explanations in this section seem to be missing?

---

## Author Comment (AC1) · 19 Feb 2021

**Answers to the comments of Vanessa Round**

Thank you for your comments, which will help us to improve our manuscript. In the following, we will give answers to all major questions and comments. The corrections and comments which are not listed here will be fully adopted in the revised manuscript.

**L 6-7:** *From the results presented it doesn't seem like there is enough evidence to state that there is similar seasonality during the quiescent phases. As you say in L261, there could potentially be seasonality present during the quiescent phases, but it could not be identified due to the limited accuracy of the Landsat velocity fields.*
**Response:** We will remove the second part of the sentence starting with 'and, to a much lesser extent …'

**L 7:** *The choice of the word 'peculiar' suggests to me that the seasonal amplitude during surging is something observed only at Harald Moltke Brae, which is not the case – perhaps 'significant', 'noteworthy', 'interesting' or something along those lines would avoid this potential misunderstanding.*
**Response:** We used the word 'peculiar' to emphasize that such a high seasonal amplitude with velocities decreasing to the level of the quiescent phases every year has not been observed before at Harald Moltke Brae. However, we agree that another word would be more appropriate here. We suggest replacing the word 'peculiar' by 'remarkable'.

**L 26-27 and Figure 1:** *It is stated that velocity remained low from the end of 2019 to the beginning of 2020 but from Figure 1 it looks like the low velocities extend to mid/late 2020 although it is hard to see for sure when the last data point is. Could the text be updated to state when (which month) in 2020 the data extends until? It would be interesting to know whether the velocity remained low over the summer 2020.*
**Response:** The velocity data in Figure 1 extends until July 2020. We will include the exact time period in the Figure caption as follows: "Flow velocity for the time period from January 1998 to July 2020 derived (…)"

**L 89:** *Is it possible that the Sentinel and TerraSAR-X data can be used to get velocity fields with temporal resolution finer than monthly, given that the time basis and temporal resolution is less than a month? If this is the case, the use of a monthly averaged dataset could mean temporal resolution is being lost. I think that the use of a monthly combined velocity dataset is reasonable, but have you analysed the individual datasets to make sure you are not losing potentially valuable information on a finer temporal scale, particularly around the rapid summer decelerations? When looking at the surge of Kyagar Glacier (doi:10.5194/tc-11-723-2017) we used TerraSAR-X data to compare velocities over consecutive 11 day periods to identify similar rapid summer decelerations.*
**Response:** Besides the monthly velocities we also computed semi-monthly velocities. In Figure 8, the velocities are shown with a semi-monthly resolution. In view of the sparsely available data before 2013, we decided to show the entire time series from 1998 to 2020 with a resolution of one month, so that it is consistent over the entire time period. Figure 1 contains all available velocities. The time series in Figure 1 and the semi-monthly velocities in Figure 8 show that the major temporal variations are already captured with a monthly time series. Instead of changing the resolution we suggest to include an additional sentence in the paper referring to the Figure 1 and stating that a higher resolution will not change our main conclusions.

**Table 1:** *Could you provide a specific range of days for the Landsat temporal resolution rather than 'few days'?*
**Response:** The Landsat velocities were estimated from various different combinations of images. This leads to large variations in the resulting time differences between consecutive velocity fields. We believe that the term 'temporal resolution' might be a bit misleading here. Therefore, we decided to replace this term by 'time difference' in the revised version of the manuscript. We will also add the following note in the caption of the table: 'Time difference denotes the interval between two consecutive velocity fields.' In the Table 1, we will show for Landsat the range from the shortest to the longest time difference (1-16 days), without considering the larger data gaps before 2000 and during winter.

**L99:** *It states that 4 DEMs are used but only three seem to be referred to – the ActicDEM and two interferometric DEMs from January 2011 and Dec/Jan 2013/2014. Is there something missing, or is the fourth DEM referring to the computed ice-height change rates?*
**Response:** It is true, that only three DEMs were used in this study. This will be corrected.

**Figure 3:** *The shading representing the most recent surge seems to cover less than the six years of reported surging. Is there a reason for this or is it an oversight? Additionally, I think it might be clearer to remove the black line joining the observed front position points, to better represent the discontinuous nature of the historical dataset.*
**Response:** The shading was originally done based on reported glacier surges prior to our study. We agree that marking the entire 6-years period of the last surge would be more appropriate here. Thus, we will extend the shading until the year 2019 which marks the end of the latest surge.

**Figure 6:** *Is there a reason not to include 2019 or even 2020 in this figure considering that the surge continued through 2019? It would give a more complete picture of this very recent surge if the data extended as far as possible, including termination of the surge in late 2019 or in 2020. I also recommend repeating the description of parameters shown on panels rather than refer back to Figure 4 – this allows Figure 6 to be understood in isolation.*
**Response**: The main data processing for this study was based on data of the front line until 2018 and climate data until 2017. In Figure 6 we confine the time to the period where the different data (climate data, frontal positions) overlap. In the further progress of our work, we only continued with the analysis of the flow velocity. The continuation of the flow velocity until 2020 is shown in Figure 1.

**Figure 7:** *The colours in the flow velocity profile plots are difficult to distinguish, especially 2013-09 and 2013-12. Please consider using a range with more contrast in these types of figures (e.g. also Figure 14, 15) or perhaps even different line patterns.*
**Response:** We will change the color map to rainbow

**Figure 8:** *The year 2018 is almost impossible to detect in the bottom panel. Could you find a way to make it visible or if it is hidden behind one of the other years perhaps make a note of this in the caption.*
**Response:** There is no data of the meltwater runoff for 2018 as the used climate data set extends only until 2017. We will add a legend in Figure 8b which does not include 2018 so that it becomes clear that there is no plot for that year.

**L 209:** *I assume this is referring to a lack of significant year-to-year changes in the lake, however it could be useful to provide a brief description of the seasonal patterns observed (lake formation in summer). In section 4.2 it is noted that large meltwater lakes form at the beginning of summer so it would be useful to make note of that here in the results section*

**Response:** The statement about the stationary lake will be included in the previous section as 'By contrast, the stationary lake at the northern side of the glacier does not exhibit any significant change visible in the Landsat images'. In addition, we will use the term 'stationary lake' already in the introduction to make the distinction between this lake and the supraglacial meltwater lakes clearer.

**Figure 11:** *I suggest considering switching the axes in this figure i.e. stacking the years vertically, to make it easier to compare the timing of events between the years (as described from L 206 onwards).*

**Response:** We agree that switching the axes can improve the readability of this plot. Therefore, we suggest plotting the years below each other so that same months will be vertically aligned and can be better compared. We will use four different colors to distinguish between 'Sea ice', 'Lakes', 'Plume' and 'Calving'. Different color saturations will indicate whether a feature is not visible, moderate or strong.

**L 275:** *It would be nice to briefly summarise the glacier types 2 and 3 identified by Moon et al. here.*

**Response:** Short explanations of the types 1, 2 and 3 of Moon will be included in the introduction of the revised manuscript as follows: "Pattern 1 exhibits slow velocities in spring with a rapid acceleration of ice flow in summer and velocities remaining high in autumn. Moon et al. (2014) explain this pattern by the seasonally changed glacier front positions. Pattern 2 is characterized by low velocities over most of the year except for a short-lasting velocity peak in mid summer. Pattern 3 shows already high velocities over several months in spring followed by a rapid deceleration in mid summer and velocities remaining low over the rest of the year. In contrast to pattern 1, the patterns 2 and 3 are assumed to be caused by the seasonally changing meltwater availability (Moon et al., 2014)." In Section 4.2 we will refer to these explanations.

**L 282:** *The observation of seasonal meltwater lakes was not noted in the results and should perhaps be included in section 3.5 too.*

**Response:** The following sentence will be added in section 3.5: "Supraglacial lakes always formed in summer followed by the formation of meltwater plumes at the glacier front."

**L 307:** *Regarding the low velocities at the beginning of 2020 – when is the latest available velocity data, i.e. from which month in 2020 and in particular do they extend into the spring/summer?*

**Response:** The exact time period for the velocities used in this study (January 1998 to July 2020) will be included in the caption of Figure 1 as well as in the introduction.

**L 310:** *The summary sentence here seems a bit subjective and also contradicts the statement in the abstract that the most recent surge 'lasted significantly longer' than previously observed surges or the passage from L 74-76. If not in terms of maximum velocity then at least in terms of surge duration the most recent surge does seem (in my subjective opinion) significantly different to at least the two well observed surges before it. The various arguments presented in section 4.3 mostly relate to limitations in the data (e.g. lack of data pre 2013, greater smoothing on velocity maxima in the earlier Landsat data). So in summary, it would be more correct*

*to say there is insufficient evidence (historical data) to conclude whether the surge behaviour has changed since 2000.*

**Response:** Instead of the sentence "In summary, …" starting in line 310 we will include the following sentence after describing the surge behavior of 1999/2000 and 2005/2006 in line 307: "However, in terms of its duration, the most recent surge clearly differs from the surges 1999/2000 and 2005/2006". And as a summary of the surge behavior of 1926-1938 and 1954-1956 the revised paper will include in line 311 the sentence: "There is not enough data to state whether there was a significant change in the surge behavior in 2000."

**L 319:** *The example of September 2013 doesn't show the moderately higher velocities further upstream which is described for pattern A, but rather seems to show velocity decreases up the glacier. Is it just the high velocity at the terminus which is the defining feature of pattern B?*

**Response:** To clarify this, we will change the description of pattern B as follows: "In pattern B, the glacier exhibits low flow velocities (<0.5 m/s) over most of its area except for a small part at the front where the velocities exceed 1 m/s."

**L 326:** *I like this concept of categorising the various flow profiles but the difference between C and D isn't very clearly defined – both of these patterns show high velocities with the maximum at the glacier front. Is the difference that with C the velocities high up the glacier are lower and hence the overall profile steeper, or that velocities are higher overall for D? Also it might be helpful to add that while pattern D reverses A, the timescales will be quite different because of the difference in the magnitude of velocity between these two patterns.*

**Response:** We will clarify this by adding the following explanations in the revised version of the paper: "Close to the glacier front the velocities are high in both cases C and D. The difference is that in pattern C the velocities are still low in the upper part of the glacier. Thus, C involves a much steeper velocity profile and a much faster surface thinning in the lower and the middle part than D. Pattern D reversed pattern A, however, with a difference in magnitude. Thus, the effect of the longer lasting quiescent phase can be compensated by a shorter lasting active phase."

---

## Author Comment (AC2) · 19 Feb 2021

**Answers to Anonymous Referee #2**

Thanks for your comments, which will help us to improve our manuscript. In the following, we will give answers to all major questions and comments. The corrections and comments which are not listed here will be fully adopted in the revised manuscript.

**L19-L41:** *I find the Introduction quite strange as it opens with a presentation of the results and study area, before providing any of the background or methods that would usually be expected in a paper. I therefore suggest moving the text from L19-L27 and Figure 1 to the Results, and the text from L28-L41 to a new section called 'Study Area' after the Introduction. The Introduction would then start on L42, although this first sentence might need to be modified.*

**Response:** We would like to start with attracting the reader by a first glance on the remarkable velocity variation patterns addressed by this study. (In fact, this was the starting point of our study.) The following paragraphs then provide a complete introduction in a conventional order. We would prefer to keep this principal structure. However, we will shorten the first paragraph to a minimum by shifting all background and explanation to the following paragraphs. Thus, our suggestion for the first and the 3rd paragraph would be as follows:

Paragraph 1:

"Based on optical and radar remote sensing data, we observed remarkable flow velocity variations of Harald Moltke Bræ, a marine-terminating outlet glacier in north-west Greenland. Figure 1 gives an impression of the variations in time, observed at a fixed position close to the terminus. Ice flow accelerated significantly in 1999/2000 and 2005/2006 and in 2013-2019. During the 2013-2019 phase, the dense temporal sampling reveals pronounced seasonal velocity variations, by one order of magnitude. At the end of 2019 velocities returned to a very low level that sustained at least until July 2020. This paper investigates in detail the spatio-temporal variations in glacier flow and geometry underlying this variation pattern."

Paragraph 3:

"Already prior to the era of satellite remote sensing, significant changes in the dynamic behaviour of Harald Moltke Bræ were reported. Wright (1939) observed an exceptional advance of the glacier front by about 2 km between 1926 and 1928 and inferred that the average surface flow velocity in this interval was at least 1000 m/year (2.7 m/day). Mock (1966) used the displacement of ice-surface features visible in aerial and terrestrial photographs to show that between 1954 and 1956 the average velocity was about 1 m/day, ten times higher than the average velocity between 1937 and 1938. Based on satellite remote sensing, the accelerated phases in 1999/2000 and 2005/2006 were previously documented by Joughin et al. (2010) and Rosenau (2014) and accelerated flow in 2013/2014 was reported by Hill et al. (2018)."

**L44-45:** *more recent papers and reviews suggest that the length of the active and quiescent phases of surge-type glaciers can be longer than what you state. For example, Jiskoot (2015) states that the quiescent phase lasts for 10s to 100s of years, while the active phase lasts for 10-15 years. See: Jiskoot H. (2011) Glacier Surging. In: Singh V.P., Singh P., Haritashya U.K. (eds) Encyclopedia of Snow, Ice and Glaciers. Encyclopedia of Earth Sciences Series, pp 415 428. Springer, Dordrecht. https://doi.org/10.1007/978-90-481-2642-2_559.*
**Response:** Thanks for mentioning this more recent publication. We will change our statement about the length of the quiescent and the active phases according to Jiskoot (2011). This reference will be added in the revised manuscript.

**L51:** *The study of Monacobreen by Murray et al. (2003) is one of the first to have reported an upglacier propagating surge, so I think that it should be referenced here.*
**Response:** We will include this reference.

**L56:** *you should also mention the large surge cluster in east Greenland, reported by studies such as Sevestre and Benn (2015) and others*
**Response:** We will include this information about clusters of surge glaciers in Greenland at line 56 as follows: "Some surge glaciers are also known in Greenland. For example, clusters of surge glaciers in central-west Greenland and central-east Greenland have been reported by Sevestre and Benn (2015). Apart from these clusters, also Hagen Brae in north-east Greenland is assumed to be a surge glacier (Solgaard et al., 2020)."

**L61:** *You need some words to introduce this sentence, such as: 'In mechanism (A), the base of a polythermal glacier: : :'. Similar to introduce (B) on L63*
**Response**: The suggested introduction of menachism (A) "In mechanism (A), the base of a polythermal glacier (…)" will be adopted in the revised manuscript."

**L78:** *I would delete 'extraordinary' as several other studies have previously documented surges initiated at the glacier terminus (e.g., see comment for L51)*
**Response:** With the word 'extraordinary' we meant to emphasize our findings about the pronounced seasonality with velocities decreasing to the level of the quiescent phase during the surge. However, as this expression could be misleading here, we propose to either remove it or use the word 'remarkably' instead.

**L85:** *I don't understand what 'suited' refers to here. Perhaps you mean 'suitable', but in that case you need to describe why the images would be suitable. In this section also make it clear that the black triangle in Fig. 2 is the point that all velocity time series were derived for – it took me a long time to spot this information in the figure caption.*
**Response**: "suited" is not important in this sentence and will be removed. We will add the following sentence in this section: "All velocity time series in this paper refer to the position indicated by the black triangle shown in Fig. 2."

**L89:** *would be useful to provide some numbers here to define what you mean by 'spatial and temporal coverage as high as possible'. e.g., max resolution, max temporal coverage Table 1 caption: change 'Overview over …' to 'Overview of …'*
**Response:** "as high as possible" means here that the spatial and temporal coverage was adopted and varies strongly depending on the characteristics and availability of different

data sets. We will remove this part of the sentence "with a spatial (…)" as this does not affect our main message which is that we computed monthly velocity fields.

**L99:** *You state that you use four different DEMs, but only list three*
**Response:** Yes, it is true that only three DEMs ware used. This will be corrected.

**L112:** *it would be useful to provide a reference or two for the choice of 0.9 to convert the surface velocity to depth-average velocity. For example, Cuffey and Paterson (2010) provide a discussion of this: Cuffey, K.M. and Paterson, W.S.B., 2010. The Physics of Glaciers. Academic Press*
**Response:** The references for the value 0.9 are given one sentence later (Wu and Jezek, 2004). Nevertheless, we will additionally include the reference Cuffey and Paterson (2010) in line L112 as suggested.

**L119-122:** *provide the resolution of your basal topography data*
**Response:** We will add at the end the sentence in line 119 "(…) gridded format with a spatial resolution of 150 m".

**L125:** *in the caption for Fig. 4 it's stated that the SMB is calculated from the difference between precipitation and runoff (i.e., it's not calculated independently), so that should be made clear here. This also seems to implicitly assume that no mass is lost from calving at the terminus, but from Fig. 5 and 7c it looks as if this could be important. Can you therefore address whether this is accounted for, and what implications this has for your SMB data? This is partly discussed in Section 3.6, but it's unclear whether your SMB values are adjusted for the calving flux.*
**Response:** The note "difference between precipitation and runoff" intends to explain the SMB, but it does not mean that we computed it in this way. We will remove this statement in the brackets. In the entire paper, we always considered a fixed area of the glacier for computing the mass balance. We defined this area to be the part of the drainage basin above a cross section, which is located close to the glacier front, but remains behind the front over the entire study period. The amount of ice lost by calving is not considered in this paper. As we focused in our study on the glacier dynamics, we are mainly interested in the deviations from the balance ice flow and corresponding to that in the ice mass and volume changes within a fixed area. For the contribution of the glacier to the sea level for example, we would need to consider the mass-balance of the entire glacier, but this was not our intention here. To clarify this, we will add a sentence in the revised paper pointing out that we actually do not consider the mass-balance of the entire glacier, but only of sub-area with a fixed size.

**L127:** *would be useful if you can show the location of this cross section on Fig. 2*
**Response:** To show the exact location of this cross section, we will include a line in Figure 2. We chose this line such that it is located close to the glacier front, but remains behind the glacier front over the entire observation period.

**L134:** *This seems to be the Results section, so it would be good to include that in the title*
**Response:** We will change the title of section 3 to "Results" replacing this long title.

**L152:** *change 'exceptional' to 'exceptionally'. Can you also talk about whether this high precipitation fell as rain or snow, and what time during the year it fell? For example, high rainfall*

*in the summer might have a different impact on dynamics compared to high snowfall in the winter.*

**Response:** We considered here only the yearly averaged data. The changes of these climatic influences were, unfortunately, not examined with a higher temporal resolution in the scope of this paper.

**Figures 4 & 6:** *please use letters (a), (b) and (c) to label and refer to the different figure parts, consistent with the other figures. In the figure caption, the text '(Black triangle in Fig. 1)' should refer to Fig. 2. It would also be useful to see the surge periods labelled on these figures (e.g., by lightly shading the background), similar to what you show in Fig. 3, so that it's clear as to how their stated start and end times of the surges match up with the velocities and frontal positions.*

**Response:** Yes, it is true that the black triangle is in Fig. 2, not in Figure 1. The references in the captions of Figure 4 & 6 will be changed accordingly! We will use (a), (b), … to denote subfigures as suggested. Shadows marking the surge periods will be added to these Figures.

**Figure 8a** *seems to show the velocities at quite a bit higher temporal resolution than in Fig. 6. The better temporal resolution allows the variations to be more clearly seen, particularly for years such as 2014 and 2015, so can you plot the full high resolution dataset (and associated errors) in Fig. 6?*

**Response:** Besides the monthly velocities we also computed semi-monthly velocities which are shown in Figure 8 for instance. In view of the sparsely available data before 2013, we decided to show the entire time series from 1998 to 2020 with a resolution of one month, so that it is consistent over the entire time period. Figure 1 contains all available velocities. The time series in Figure 1 and the semi-monthly velocities in Figure 8 show that the major temporal variations are already contained in the monthly time series. Instead of changing the resolution we suggest to include an additional sentence in the paper referring to Figure 1 and stating that a higher resolution will not change our main conclusions.

**L214-220:** *this discussion only refers to mass lost by flow through the cross-section close to the glacier front. However, mass is also lost by retreat of the glacier terminus (and mass gained when it advances), so this should also be accounted for when presenting the mass numbers and balancing them against the flux through the crosssection and the SMB.*

**Response:** Here, we consider only the mass balance of the glacier area above this cross section (see also the more detailed response to the comment of line 125).

**Figure 11:** *there is insufficient information provided in the figure caption or methods to understand what exactly was measured for these visual features, and how a distinction is made between 'moderate' and 'strong' conditions. The methods refer to Appendix E, but this simply consists of three satellite images without any description. Please provide more information so that the reader can understand what was done, and what was measured.*

**Response:** We will include an explanation of the classification of the features in Appendix E based on examples in the selected Landsat scenes. We well refer to Appendix E in the Figure caption.

**L252:** *at the start of this section I would like to see a few sentences to describe exactly how you defined a surge, including its start and end date. For example, is this just based on velocity variations, or also on things such as changes in terminus position, calving rate, surface cre-*

*vassing, etc.? This is needed to put the remainder of this section in perspective, particularly because the 2013-19 surge was so much longer than the previous two surges.*

**Response:** We will add the following sentence at the beginning of this section: "In this context as the main criterion for the surge the strong change of flow velocity is used, as discussed in the introduction."

**L266-273:** *here and elsewhere in this section it would be useful to make better reference to the figures*

**Response:** We agree that references to the figures are missing in this section. The statements are mainly based on Figure 8. We will add the references to this figure.

**L283:** *please provide some more detail about the large meltwater lakes. E.g., are these supraglacial? Ice marginal? Where on the glacier do they form?*

**Response:** We will mention in the text, that we refer to the supraglacial lakes here. As an example of these supraglacial lakes we will refer to Appendix E.

**L322:** *can you provide the maximum velocity for the third pattern?*

**Response:** The maximum velocities will be included in the explanation as follows: "… with a maximum of about 6 - 10 m/yr… "

**L316-337:** *I find it a bit hard to follow and visualize the flow patterns based on the text descriptions here. To make them easier to understand, could you add a schematic or cartoon that shows the velocities and surface elevation changes associated with each pattern (A, B, C, D), and the sequence that occurs during surge initiation (A-B-C-D) and after surge initiation (A-D)? This will also help with the description of upglacier propagation vs uniform acceleration described in Section 4.5*

**Response:** We will add an additional Figure showing schematically one ideal profile for each of the flow velocity patterns and the 4 corresponding patterns of the ice height change.

**L343:** *can you provide a reference to support the statement that 'Surge looped moraines typically indicate a down-glacier surge propagation'?*

**Response:** This is actually an assumption for which we do not have any reference. We will remove this statement, if we cannot find and reference for it.

**L353:** *would also be useful to make comparisons with other up-glacier propagating surges, as well as the findings of this paper (particularly for Section 4.7): Thøgersen, K. et al. 2019. Rate-and-state friction explains glacier surge propagation. Nature Communications, 10, 2823.*

**Response:** We actually made some comparisons with an up-glacier spreading of a surge in Svalbard observed by (Sevestre et al. 2018). However, Thøgersen, K. et al. (2019) will be included as an additional reference.

**L366:** *can you provide more information about this? E.g., what are the mean annual temperatures here? Have there been any direct measurements or evidence of polythermal conditions at this glacier or other nearby ones of a similar size?*

**Response:** We decided to delete the description and arguments of the polythermal glacier in the lines 366-368 as we do not have enough evidence for that. Instead, we will begin with the hydrological argument, and after that shortly explain why a thermal mechanism is unlikely.

**L375:** *please describe what the 'Kamb drainage-switching theory' is for any readers who might not be familiar with it.*
**Response:** We will delete the sentence with the Kamb drainage switch theory as this theory is not that relevant for our studies at Harald-Moltke-Brae.

**L389-394:** *do you have any observations of crevasse formation at Harald Moltke Brae from the remote sensing imagery that provide support for the statements made here? It could be useful to add them as a visual feature to Fig. 11*
**Response:** The resolution of the applied Landsat images is too low for a detailed analysis of the crevasses.

**L421:** *it would be useful to provide some numbers here for the mass imbalance, including the mass due to terminus retreat (related to comment for L214-220)*
**Response:** Our mass balance considerations refer always to fixed area (see explanations for L214-220).

**L431:** *was any change in crevasse formation actually observed (similar comment to L389-394)? It could also be useful to refer to the visual indicators from Fig. 11 here or elsewhere in the Conclusions to help back up your statements.*
**Response:** see answer to comment for lines 389-394.

**L450:** *I don't understand the reference to improvement in spatial resolution here, when both Landsat 7 and 8 have a multispectral resolution of 30 m and panchromatic resolution of 15 m*
**Response:** The resolution refers here to the computed velocity fields (Rosenau, 2014). This statement will be added in the text for clarification.

**Appendix E:** *text explanations in this section seem to be missing?*
**Response:** Some more explanation will be added here describing how we distinguish between low, moderate and strong occurrence of a feature based on these example images.

---

## Author Response (AR1)

Thank you again for your valuable comments and corrections. Below we provide answers to all major questions and comments. They are essentially the same as posted in the discussion, but slightly adapted to the changes we have finally made. Any corrections not listed here have been fully incorporated into the revised manuscript. In addition, some minor corrections from our side are also listed below.

**Answers to the comments of Vanessa Round**

**L 6-7:** From the results presented it doesn't seem like there is enough evidence to state that there is similar seasonality during the quiescent phases. As you say in L261, there could potentially be seasonality present during the quiescent phases, but it could not be identified due to the limited accuracy of the Landsat velocity fields.

**Response:** We have removed the second part of the sentence starting with 'and, to a much lesser extent ...'

**L 7:** The choice of the word 'peculiar' suggests to me that the seasonal amplitude during surging is something observed only at Harald Moltke Brae, which is not the case – perhaps 'significant', 'noteworthy', 'interesting' or something along those lines would avoid this potential misunderstanding.

**Response:** We used the word 'peculiar' to emphasize that such a high seasonal amplitude with velocities decreasing to the level of the quiescent phases every year has not been observed before at Harald Moltke Brae. However, we agree that another word would be more appropriate here. We have replaced the word 'peculiar' by 'remarkable'.

**L 26-27 and Figure 1:** It is stated that velocity remained low from the end of 2019 to the beginning of 2020 but from Figure 1 it looks like the low velocities extend to mid/late 2020 although it is hard to see for sure when the last data point is. Could the text be updated to state when (which month) in 2020 the data extends until? It would be interesting to know whether the velocity remained low over the summer 2020.

**Response:** The velocity data in Figure 1 extends until July 2020. We have included the exact time period in the Figure caption as follows: "Flow velocity for the time period from January 1998 to July 2020 derived (...)"

**L 89:** Is it possible that the Sentinel and TerraSAR-X data can be used to get velocity fields with temporal resolution finer than monthly, given that the time basis and temporal resolution is less than a month? If this is the case, the use of a monthly averaged dataset could mean temporal resolution is being lost. I think that the use of a monthly combined velocity dataset is reasonable, but have you analysed the individual datasets to make sure you are not losing potentially valuable information on a finer temporal scale, particularly around the rapid summer decelerations? When looking at the surge of Kyagar Glacier (doi:10.5194/tc-11-723-2017) we used TerraSAR-X data to compare velocities over consecutive 11 day periods to identify similar rapid summer decelerations.

**Response:** Besides the monthly velocities we also computed semi-monthly velocities. In Figure 8, the velocities are shown with a semi-monthly resolution. In view of the sparsely available data before 2013, we decided to show the entire time series from 1998 to 2020 with a resolution of one month, so that it is consistent over the entire time period. Figure 1 contains all available velocities. The time series in Figure 1 and the semi-monthly velocities in Figure 8 show that the major temporal variations are already captured with a monthly time series. Instead of

changing the resolution we included an additional sentence in the paper referring to the Figure 1 and stating that a higher resolution will not change our main conclusions.

**Table 1:** Could you provide a specific range of days for the Landsat temporal resolution ratherthan 'few days'?**

**Response:** The Landsat velocities were estimated from various different combinations of images. This leads to large variations in the resulting time differences between consecutive velocity fields. We believe that the term 'temporal resolution' might be a bit misleading here. Therefore, we decided to replace this term by 'time difference' in the revised version of the manuscript. We added the following note in the caption of the table: 'The time difference denotes the interval between two consecutive velocity fields.' In the Table 1, we show for Landsat the range from the shortest to the longest time difference (1-16 days), without considering the larger data gaps before 2000 and during winter.

**L99:** It states that 4 DEMs are used but only three seem to be referred to – the ActicDEM and two interferometric DEMs from January 2011 and Dec/Jan 2013/2014. Is there something missing, or is the fourth DEM referring to the computed ice-height change rates? **Response:** It is true, that only three DEMs were used in this study. This has been corrected.

**Figure 3:** The shading representing the most recent surge seems to cover less than the six years of reported surging. Is there a reason for this or is it an oversight? Additionally, I think it might be clearer to remove the black line joining the observed front position points, to better represent the discontinuous nature of the historical dataset.

**Response:** The shading was originally done based on reported glacier surges prior to our study. We agree that marking the entire 6-years period of the last surge would be more appropriate here. Thus, we extended the shading until the year 2019 which marks the end of the latest surge.

**Figure 6:** Is there a reason not to include 2019 or even 2020 in this figure considering that the surge continued through 2019? It would give a more complete picture of this very recent surge if the data extended as far as possible, including termination of the surge in late 2019 or in 2020. I also recommend repeating the description of parameters shown on panels rather than refer back to Figure 4 – this allows Figure 6 to be understood in isolation.

**Response**: The main data processing for this study was based on data of the front line until 2018 and climate data until 2017. In Figure 6 we confine the time to the period where the different data (climate data, frontal positions) overlap. In the further progress of our work, we only continued with the analysis of the flow velocity. The continuation of the flow velocity until 2020 is shown in Figure 1.

**Figure 7:** The colours in the flow velocity profile plots are difficult to distinguish, especially 2013-09 and 2013-12. Please consider using a range with more contrast in these types of figures (e.g. also Figure 14, 15) or perhaps even different line patterns. **Response:** We have changed the color map of these Figures to 'jet'

**Figure 8:** The year 2018 is almost impossible to detect in the bottom panel. Could you find a way to make it visible or if it is hidden behind one of the other years perhaps make a note of this in the caption.

**Response:** There is no data of the meltwater runoff for 2018 as the used climate data set extends only until 2017. We have added a legend in Figure 8b which does not include 2018 so that it becomes clear that there is no plot for that year.

**L 209:** I assume this is referring to a lack of significant year-to-year changes in the lake, however it could be useful to provide a brief description of the seasonal patterns observed (lake formation in summer). In section 4.2 it is noted that large meltwater lakes form at the beginning of summer so it would be useful to make note of that here in the results section

**Response:** The statement about the stationary lake was included in the previous section as 'In contrast to the supraglacial lakes, the stationary lake at the northern side of the glacier does not exhibit any significant seasonal or long-term variations visible in the Landsat images'. In addition, we now use the term 'stationary lake' already in the introduction to make the distinction between this lake and the supraglacial meltwater lakes clearer.

**Figure 11:** I suggest considering switching the axes in this figure i.e. stacking the years vertically, to make it easier to compare the timing of events between the years (as described from L 206 onwards).

**Response:** We agree that switching the axes can improve the readability of this plot. Therefore, we now plotted the years below each other so that same months will be vertically aligned and can be better compared. We will use four different colors to distinguish between 'Sea ice', 'Lakes', 'Plume' and 'Calving'. Grey colour, hatch fill and solid fill indicate whether a feature is not visible, moderate or strong, respectively.

**Figure 14:** It would be helpful to show a horizontal line at 0 to make it easier to distinguish between positive/negative height gain (especially for the earlier dates for where the difference is more subtle). The same applies to Figure 15.

We have added a solid line at 0 in Fig. 14 and 15.

**L 275:** It would be nice to briefly summarise the glacier types 2 and 3 identified by Moon et al. *here.*

**Response:** Short explanations of the types 1, 2 and 3 of Moon are now included in the introduction.

**L 282:** *The observation of seasonal meltwater lakes was not noted in the results and should perhaps be included in section 3.5 too.*

**Response:** The following sentence has been added in section 3.5: "Supraglacial lakes always formed in summer followed by the formation of meltwater plumes at the glacier front."

**L 307:** Regarding the low velocities at the beginning of 2020 – when is the latest available velocity data, i.e. from which month in 2020 and in particular do they extend into the spring/summer?

**Response:** The exact time period for the velocities used in this study (January 1998 to July 2020) is now included in the caption of Figure 1 as well as in the introduction.

**L 310:** The summary sentence here seems a bit subjective and also contradicts the statement in the abstract that the most recent surge 'lasted significantly longer' than previously observed surges or the passage from L 74-76. If not in terms of maximum velocity then at least in terms of surge duration the most recent surge does seem (in my subjective opinion) significantly different to at least the two well observed surges before it. The various arguments presented in section 4.3 mostly relate to limitations in the data (e.g. lack of data pre 2013, greater smoothing on velocity maxima in the earlier Landsat data). So in summary, it would be more

correct to say there is insufficient evidence (historical data) to conclude whether the surge behaviour has changed since 2000.

**Response:** Instead of the sentence "In summary, …" starting in line 310 we have included the following sentence after describing the surge behavior of 1999/2000 and 2005/2006 in line 307: "However, in terms of its duration, the most recent surge clearly differs from the surges 1999/2000 and 2005/2006". And as a summary of the surge behavior of 1926-1938 and 1954-1956 the revised paper includes the following sentence at the end of the paragraph 4.3: "In summary, there is not enough data to state whether the surges 1926-1928 and 1954-1956 were significantly different from the more recent surges."

**L 319:** The example of September 2013 doesn't show the moderately higher velocities further upstream which is described for pattern A, but rather seems to show velocity decreases up the glacier. Is it just the high velocity at the terminus which is the defining feature of pattern B?

**Response:** To clarify this, we changed the description of pattern B as follows: "In pattern B, the glacier exhibits low flow velocities (<0.5 m/day) over most of its area except for a small part at the front where the velocities exceed 1 m/day."

**L 326:** I like this concept of categorising the various flow profles but the difference between C and D isn't very clearly defined – both of these patterns show high velocities with the maximum at the glacier front. Is the difference that with C the velocities high up the glacier are lower and hence the overall profile steeper, or that velocities are higher overall for D? Also it might be helpful to add that while pattern D reverses A, the timescales will be quite different because of the difference in the magnitude of velocity between these two patterns.

**Response:** We clarify this by adding the following explanations in the revised version of the paper: "(D) The fourth pattern shows high velocities at the glacier front, similar to pattern (C). (D) differs from (C) in that it has higher velocities in the middle and the upper part of the glacier (Fig. 16). This is associated with a more gently sloped velocity profile in the lower10 km of the glacier. As a consequence, the glacier is dynamically thickening in its lower part and thinning in its upper part. Pattern (D) reverses the combined effect of the patterns (A) and (B), however, with a difference in magnitude. Thus, the effect of the longer lasting quiescent phase can be compensated by a shorter lasting active phase"

**L 227:** There is a typo on the stated depth of 230 km (should be m). L 251: Typo 'glaclier' rather than glacier. Corrected!

**Answers to Anonymous Referee #2**

**L19-L41:** I find the Introduction quite strange as it opens with a presentation of the results and study area, before providing any of the background or methods that would usually be expected in a paper. I therefore suggest moving the text from L19-L27 and Figure 1 to the Results, and the text from L28-L41 to a new section called 'Study Area' after the Introduction. The Introduction would then start on L42, although this first sentence might need to be modified.

**Response:** We would like to start with attracting the reader by a first glance on the remarkable velocity variation patterns addressed by this study. (In fact, this was the starting point of our study.) The following paragraphs then provide a complete introduction in a conventional order. We would prefer to keep this principal structure. However, we shortened the first paragraph to a minimum by shifting all background and explanation to the following paragraphs. Thus, our suggestion for the first and the 3rd paragraph would be as follows:

**Paragraph 1:**

"Based on optical and radar remote sensing data, we observed remarkable flow velocity variations of Harald Moltke Bræ, a marine-terminating outlet glacier in north-west Greenland. Figure 1 gives an impression of the variations in time, observed at a fixed position close to the terminus. Ice flow accelerated significantly in 1999/2000 and 2005/2006 and in 2013-2019. During the 2013-2019 phase, the dense temporal sampling reveals pronounced seasonal velocity variations, by one order of magnitude. At the end of 2019, velocities returned to a very low level that sustained at least until July 2020. This paper investigates in detail the spatio-temporal variations in glacier flow and geometry underlying this variation pattern."

**Paragraph 3:**

"Already prior to the era of satellite remote sensing, significant changes in the dynamic behaviour of Harald Moltke Bræ were reported. Wright (1939) observed an exceptional advance of the glacier front by about 2 km between 1926 and 1928 and inferred that the average surface flow velocity in this interval was at least 1000 m/year (2.7 m/day). Mock (1966) used the displacement of ice-surface features visible in aerial and terrestrial photographs to show that between 1954 and 1956 the average velocity was about 1 m/day, ten times higher than the average velocity between 1937 and 1938. Based on satellite remote sensing, the accelerated phases in 1999/2000 and 2005/2006 were previously documented by Joughin et al. (2010) and Rosenau (2014) and accelerated flow in 2013/2014 was reported by Hill et al. (2018)."

**L44-45:** more recent papers and reviews suggest that the length of the active and quiescent phases of surge-type glaciers can be longer than what you state. For example, Jiskoot (2015) states that the quiescent phase lasts for 10s to 100s of years, while the active phase lasts for 10-15 years. See: Jiskoot H. (2011) Glacier Surging. In: Singh V.P., Singh P., Haritashya U.K. (eds) Encyclopedia of Snow, Ice and Glaciers. Encyclopedia of Earth Sciences Series, pp 415 428. Springer, Dordrecht. https://doi.org/10.1007/978-90-481-2642-2\_559.

**Response:** Thanks for mentioning this more recent publication. We changed our statement about the length of the quiescent and the active phases according to Jiskoot (2011). This reference is now also included in the revised manuscript.

**L51:** The study of Monacobreen by Murray et al. (2003) is one of the first to have reported an upglacier propagating surge, so I think that it should be referenced here. **Response:** We have included this reference.

**L56:** you should also mention the large surge cluster in east Greenland, reported by studies such as Sevestre and Benn (2015) and others

**Response:** We have included this information about clusters of surge glaciers in Greenland as follows: "Some surge glaciers are also known in Greenland. For example, clusters of surge glaciers in central-west Greenland and central-east Greenland have been reported by Sevestre and Benn (2015). Apart from these clusters, also Hagen Bræ in north-east Greenland is assumed to be a surge glacier (Solgaard et al., 2020)."

**L61:** You need some words to introduce this sentence, such as: 'In mechanism (A), the base of a polythermal glacier: ::'. Similar to introduce (B) on L63

**Response**: We have changed the sentence such that is start as "Mechanism (A) occurs in glaciers which (...)" was adopted in the revised manuscript."

**L78:** I would delete 'extraordinary' as several other studies have previously documented surges initiated at the glacier terminus (e.g., see comment for L51)

**Response:** With the word 'extraordinary' we meant to emphasize our findings about the pronounced seasonality with velocities decreasing to the level of the quiescent phase during the surge. However, as this expression could be misleading here, we removed it.

**L85:** I don't understand what 'suited' refers to here. Perhaps you mean 'suitable', but in that case you need to describe why the images would be suitable. In this section also make it clear that the black triangle in Fig. 2 is the point that all velocity time series were derived for – it took me a long time to spot this information in the figure caption.

**Response**: "suited" is not important in this sentence and was removed. We added the following sentence in this section: "All velocity time series in this paper refer to the position indicated by the black triangle shown in Fig. 2."

**L89:** would be useful to provide some numbers here to define what you mean by 'spatial and temporal coverage as high as possible'. e.g., max resolution, max temporal coverage Table 1 caption: change 'Overview over ...' to 'Overview of ...'

**Response:** "as high as possible" means here that the spatial and temporal coverage was adopted and varies strongly depending on the characteristics and availability of different data sets. We have removed this part of the sentence "with a spatial (...)" as this does not affect our main message which is that we computed monthly velocity fields.

**L99:** *You state that you use four different DEMs, but only list three* **Response:** Yes, it is true that only three DEMs ware used. This has been corrected.

**L112:** *it would be useful to provide a reference or two for the choice of 0.9 to convert the surface velocity to depth-average velocity. For example, Cuffey and Paterson (2010) provide a discussion of this: Cuffey, K.M. and Paterson, W.S.B., 2010. The Physics of Glaciers. Academic Press*

**Response:** The references for the value 0.9 are given one sentence later (Wu and Jezek, 2004). Nevertheless, we have also included the reference Cuffey and Paterson (2010) in line L112 as suggested.

**L119-122: provide the resolution of your basal topography data**

**Response:** We have added at the end the sentence in line 119 "(...) gridded format with a spatial resolution of 150 m".

**L125:** in the caption for Fig. 4 it's stated that the SMB is calculated from the difference between precipitation and runoff (i.e., it's not calculated independently), so that should be made clear here. This also seems to implicitly assume that no mass is lost from calving at the terminus, but from Fig. 5 and 7c it looks as if this could be important. Can you therefore address whether this is accounted for, and what implications this has for your SMB data? This is partly discussed in Section 3.6, but it's unclear whether your SMB values are adjusted for the calving flux.

**Response:** The note "difference between precipitation and runoff" intends to explain the SMB, but it does not mean that we computed it in this way. We removed this statement in the brackets. In the entire paper, we always considered a fixed area of the glacier for computing the mass balance. We defined this area to be the part of the drainage basin above a cross section, which is located close to the glacier front, but remains behind the front over the entire study period. The amount of ice lost by calving is not considered in this paper. As we focused in our study on the glacier dynamics, we are mainly interested in the deviations from the balance ice flow and corresponding to that in the ice mass and volume changes within a fixed area. For the contribution of the glacier to the sea level for example, we would need to consider the mass-balance of the entire glacier, but this was not our intention here. To clarify this, we have added a sentence in the revised paper pointing out that we actually do not consider the massbalance of the entire glacier, but only of sub-area with a fixed size.

**L127:** would be useful if you can show the location of this cross section on Fig. 2**

**Response:** To show the exact location of this cross section, we included a line in Figure 2. We chose this line such that it is located close to the glacier front, but remains behind the glacier front over the entire observation period.

**L134:** *This seems to be the Results section, so it would be good to include that in the title* **Response:** We changed the title of section 3 to "Results" replacing this long title.

**L152:** change 'exceptional' to 'exceptionally'. Can you also talk about whether this high precipitation fell as rain or snow, and what time during the year it fell? For example, high rainfall in the summer might have a different impact on dynamics compared to high snowfall in the winter.

**Response:** We considered here only the yearly averaged data. The changes of these climatic influences were, unfortunately, not examined with a higher temporal resolution in the scope of this paper.

**Figures 4 & 6:** please use letters (a), (b) and (c) to label and refer to the different figure parts, consistent with the other figures. In the figure caption, the text '(Black triangle in Fig. 1)' should refer to Fig. 2. It would also be useful to see the surge periods labelled on these figures (e.g., by lightly shading the background), similar to what you show in Fig. 3, so that it's clear as to how their stated start and end times of the surges match up with the velocities and frontal positions.

**Response:** Yes, it is true that the black triangle is in Fig. 2, not in Figure 1. The references in the captions of Figure 4 & 6 have been changed accordingly! We now use (a), (b), ... to denote subfigures as suggested. Shadows marking the surge periods have been added to these Figures.

**Figure 8a** seems to show the velocities at quite a bit higher temporal resolution than in Fig. 6. The better temporal resolution allows the variations to be more clearly seen, particularly for years such as 2014 and 2015, so can you plot the full high resolution dataset (and associated errors) in Fig. 6?

**Response:** Besides the monthly velocities we also computed semi-monthly velocities which are shown in Figure 8 for instance. In view of the sparsely available data before 2013, we decided to show the entire time series from 1998 to 2020 with a resolution of one month, so that it is consistent over the entire time period. Figure 1 contains all available velocities. The time series in Figure 1 and the semi-monthly velocities in Figure 8 show that the major temporal variations are already contained in the monthly time series. Instead of changing the resolution we have included an additional sentence in the paper referring to Figure 1 and stating that a higher resolution will not change our main conclusions (L95).

**L214-220:** this discussion only refers to mass lost by flow through the cross-section close to the glacier front. However, mass is also lost by retreat of the glacier terminus (and mass gained when it advances), so this should also be accounted for when presenting the mass numbers and balancing them against the flux through the crosssection and the SMB.

**Response:** Here, we consider only the mass balance of the glacier area above this cross section (see also the more detailed response to the comment of line 125).

**Figure 11:** there is insufficient information provided in the figure caption or methods to understand what exactly was measured for these visual features, and how a distinction is made between 'moderate' and 'strong' conditions. The methods refer to Appendix E, but this simply consists of three satellite images without any description. Please provide more information so that the reader can understand what was done, and what was measured. **Response:** We included an explanation for the classification of the features in Appendix E based on examples in the selected Landsat scenes. We well refer to Appendix E in the caption of Figure 11.

**L252:** at the start of this section I would like to see a few sentences to describe exactly how you defined a surge, including its start and end date. For example, is this just based on velocity variations, or also on things such as changes in terminus position, calving rate, surface crevassing, etc.? This is needed to put the remainder of this section in perspective, particularly because the 2013-19 surge was so much longer than the previous two surges. **Response:** We added the following sentence in this section: "For the identification of the surges periods, as mentioned in Section 1, we use both the strong change of flow velocity and the terminus advance as main criteria"

**L266-273:** here and elsewhere in this section it would be useful to make better reference to the figures

**Response:** We agree that references to the figures are missing in this section. The statements are mainly based on Figure 8. We have added the references to this figure.

**L283:** please provide some more detail about the large meltwater lakes. E.g., are these supraglacial? Ice marginal? Where on the glacier do they form?

**Response:** We now mention in the text, that we refer to the supraglacial lakes here. As an example of these supraglacial lakes we refer to Appendix E.

**L322:** can you provide the maximum velocity for the third pattern?**

**Response:** The maximum velocities are now included in the explanation as follows: "... with a maximum of about 6 - 10 m/day..."

**L316-337:** I find it a bit hard to follow and visualize the flow patterns based on the text descriptions here. To make them easier to understand, could you add a schematic or cartoon that shows the velocities and surface elevation changes associated with each pattern (A, B, C, D), and the sequence that occurs during surge initiation (A-B-C-D) and after surge initiation (A-D)? This will also help with the description of upglacier propagation vs uniform acceleration described in Section 4.5

**Response:** We have added an additional Figure showing schematically one ideal profile for each of the flow velocity patterns and the 4 corresponding patterns of the ice height change.

**L343:** can you provide a reference to support the statement that 'Surge looped moraines typically indicate a down-glacier surge propagation'?

**Response:** This is actually an assumption for which we do not have any reference. We have removed this statement, if we cannot find and reference for it.

**L353:** would also be useful to make comparisons with other up-glacier propagating surges, as well as the findings of this paper (particularly for Section 4.7): Thøgersen, K. et al. 2019. Rate-and-state friction explains glacier surge propagation. Nature Communications, 10, 2823.

**Response:** We made some comparisons with an up-glacier spreading of a surge in Svalbard observed by (Sevestre et al. 2018). In the end we have not added the reference *Thøgersen et al. (2019) as it does not integrate so well with our explanations.*

**L366:** can you provide more information about this? E.g., what are the mean annual temperatures here? Have there been any direct measurements or evidence of polythermal conditions at this glacier or other nearby ones of a similar size?

**Response:** We decided to delete the description and arguments of the polythermal glacier in the lines 366-368 as we do not have enough evidence for that. Instead, we now begin with the hydrological argument, and after that shortly explain why a thermal mechanism is unlikely.

**L375:** please describe what the 'Kamb drainage-switching theory' is for any readers who might not be familiar with it.

**Response:** We have deleted the sentence with the Kamb drainage switch theory as this theory is not that relevant for our studies at Harald-Moltke-Brae.

**L389-394:** do you have any observations of crevasse formation at Harald Moltke Brae from the remote sensing imagery that provide support for the statements made here? It could be useful to add them as a visual feature to Fig. 11

**Response:** The resolution of the applied Landsat images is too low for a detailed analysis of the crevasses.

**L421:** it would be useful to provide some numbers here for the mass imbalance, including the mass due to terminus retreat (related to comment for L214-220)

**Response:** Our mass balance considerations refer always to fixed area (see explanations for L214-220).

**L431:** was any change in crevasse formation actually observed (similar comment to L389-394)? It could also be useful to refer to the visual indicators from Fig. 11 here or elsewhere in the Conclusions to help back up your statements.

**Response:** see answer to comment for lines 389-394.

**L450:** I don't understand the reference to improvement in spatial resolution here, when both Landsat 7 and 8 have a multispectral resolution of 30 m and panchromatic resolution of 15 m

**Response:** The resolution refers here to the computed velocity fields (Rosenau, 2014). This statement has added in the text for clarification.

**Figure A1:** there have been calls to stop using the terms 'slave-master' in remote sensing as they can be interpreted as colonial terms. Something such as 'reference' and 'secondary' is better.

**Response:** We changed the label to "Time basis [days]". The explanation for this term is given in Table 1.

**Appendix E:** text explanations in this section seem to be missing?

**Response:** We have added some text here describing how we distinguish between low, moderate, and strong occurrence of a feature based on these example images.

**Further corrections**

Besides the reviewers' comments, we have made the following changes:

L7: The sentence "It is peculiar to Harald Moltke Brae that …" has been removed since it depends on the statement in the previous sentence that seasonality was present during the quiescent phase which was deleted."

L44: We do not use the term "transition area" anymore as it could be a bit misleading here. Instead, we only talk about lower and upper part of the glacier or reservoir and receiving area.

L46: Most glacier surges started with [...] and propagated downglacier

L117: v is the **depth-averaged** velocity

L125: Bedmachine v3

L128: added "above 16 km of the center line in Fig. 2"

L247&248: Fig.13**b**

Fig. 13b: We have reversed the colormap.

L150: The cross section is actually located at about 15 km, not 18 km.

L135: The last sentence of the paragraph now correctly reads "To suppress noise, we fit a line to the ice thickness profiles and a 4th degree polynomial to the velocity profiles."

---

## Author Response (AR2)

**Answer to the editor**

Dear Daniel,

Thanks for accepting our paper for publication and giving a guideline for the last set of minor revisions.

We have adopted most of the reviewer's suggestions. In order to track the latest changes, we have uploaded a marked-up version that shows the differences between the current manuscript (version4) and the previous one (version3).

Somewhat major changes have been made to the structure of the introduction: We agree that the introduction is better structured if it first gives some background information on the topic before presenting the results. Our intention to attract the reader right at the beginning should already be achieved with the abstract. Therefore, we decided to start the introduction with some general information about surge-type glaciers, followed by a short explanation of the motivation for our study of Harald Moltke Brae. We then go into more detail about the characteristics and mechanisms of glacier surges. The time series showing the flow velocity from 1998 – 2019 has been moved to the 'Results' section.

As suggested by the reviewer, we now present the geography of the Harald Moltke Brae in a separate section entitled 'Study area'.

We have also added some more detailed explanations (comments for lines 95, 99, 127-128 and 132-133) which we hope will provide more clarity on our approaches.

Best regards
Lukas